# DISENTANGLING MULTIMODAL KNOWLEDGE PRESERVATION AND EDITING VIA LOW-RANK ADAPTATION

## ABSTRACT

Knowledge editing facilitates precise and targeted updates in Large Language Models (LLMs) and Multimodal Models (LMMs) without the need for full retraining. Although existing editing methods achieve remarkable performance in textual modality, they still struggle to simultaneously preserve pre-trained knowledge and generalize new knowledge in intricate multimodal contexts. To address this challenge, we propose ELoRA, a novel solution that disentangles the conflicting editing objectives. Specifically, ELoRA decomposes the standard Low-Rank Adaptation (LoRA) update into two complementary subspaces: (1) a null space aligned with preserved knowledge, constructed via multimodal initialization to maintain the model's general capabilities, and (2) a knowledge space extracted from the model's internal states to multimodal perturbations, capturing invariant semantics of updates. Extensive experiments on various LMMs, including LLaVA-v1.5-7B, Qwen2.5-VL-7B, and Phi-4-multimodal, show that ELoRA outperforms most LoRA-based methods by an average of 14.2% accuracy across three metrics: reliability, generality, and locality under rigorous LLM-as-a-Judge evaluation, which demonstrates that ELoRA can achieve superior real-world editing quality.

## 1 INTRODUCTION

Large Multimodal Models (LMMs) have demonstrated remarkable generality and cross-modal reasoning capabilities (Liu et al., 2023; Li et al., 2023; Qwen et al., 2025). Nevertheless, aligning their static knowledge with evolving facts and domain-specific requirements remains a fundamental challenge. To this end, knowledge editing has emerged as a promising approach for precise, targeted updates to LMMs (Cheng et al., 2023; Du et al.).

Knowledge Editing (KE) methods are broadly categorized into intrinsic parameter modification (Meng et al., 2022; Meng et al.; Deng et al., 2025; Fang et al., 2025) and external knowledge retrieval (Zheng et al.; Pan et al., 2024). External KE, while adaptable, often faces challenges in retrieval latency and relies on external databases. In contrast, intrinsic KE offers permanent updates by directly modifying model parameters, yielding deeper knowledge embedding (Zhang et al., 2024). However, existing intrinsic KE methods often compromise the retained knowledge and struggle to edit intricate multimodal knowledge, especially generalizable beyond simple factual triplets. (as shown in Figure 1b). Our work focuses on advancing intrinsic KE to address these limitations.

Parameter-Efficient Fine-Tuning (PEFT) offers an efficient means of intrinsic KE in LMMs. Recent work (Yu et al., 2024; Wang et al., 2024; Chen et al., 2025; Li et al., 2025) has successfully applied Low-Rank Adaptation (LoRA) (Hu et al.) to update both textual and visual knowledge. LoRA fine-tunes large pre-trained models by introducing a small set of low-rank matrices under next-token prediction objectives, which makes it ideal for edits beyond simple factual triplets towards more complex multimodal knowledge.

Despite their success, directly leveraging LoRA for multimodal KE faces challenges. First, the LoRA update $\Delta \mathbf{W} = \mathbf{B}\mathbf{A}$ introduces a dense matrix in addition to pre-trained weights, potentially impacting prior knowledge, and thus leading to **low locality** (Wang et al., 2024). Second, prevalent KE methods fine-tuning on a single or a few edit samples (a common issue not exclusive to LoRA) risks overfitting to sample-specific noise or low-level features, thereby hindering the acquisition of high-level semantic knowledge and intended edits on relevant samples, resulting in **low generality**.

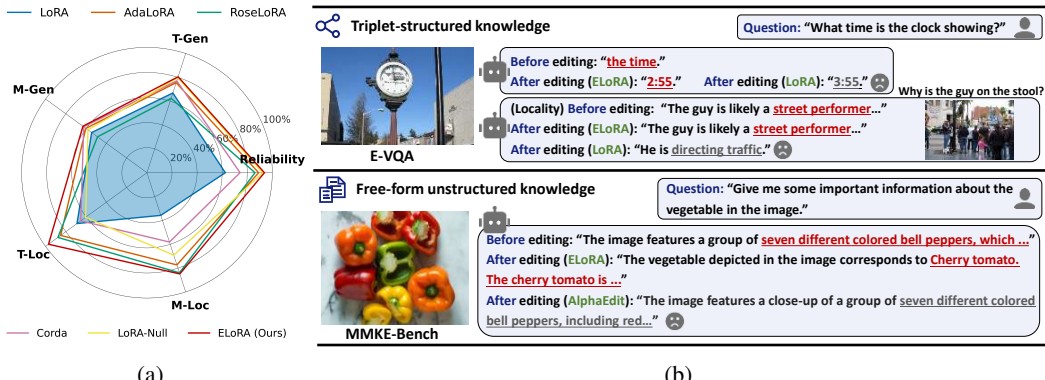

(a)                                                                                     (b)

Figure 1: (a) Comparison of editing performance for various PEFT methods on LLaVA-v1.5-7B (Liu et al., 2023). ELoRA achieves balanced superiority across reliability, generality (Gen), and locality (Loc) in both textual (T-) and visual modality (M-). (b) Editing case studies of triplet knowledge in E-VQA and free-form knowledge in MMKE-Bench. More cases are provided in Appendix H.

To mitigate the above issues, we propose **ELoRA**, a constraint editing solution based on LoRA that achieves both localized and generalizable knowledge updates. Rather than directly applying dense low-rank updates, ELoRA strategically decomposes the LoRA matrices into semantically aligned subspaces that preserve or edit the high-level semantics of the knowledge.

Specifically, to ensure locality, the model's internal states of preserved knowledge should be kept effectively unchanged. To achieve this, we project matrix $\mathbf{A}$ onto **the null subspace (Wang et al., 2021) of preserved knowledge** and keep it frozen during updates, which constrains its output for preserved knowledge inputs to near-zero, thereby preventing unintended interference.

To improve generality, updates should be performed on the shared, invariant semantics across relevant samples rather than on specific instances. To achieve this, we extract the intended edit's invariant semantics from the model's internal states across a wide distribution of perturbed inputs and construct a **generalizable yet centralized low-rank knowledge subspace**. Then we project the update of matrix $\mathbf{B}$ onto the knowledge space. The key insight is that an update from a single or a few samples is prone to overfitting superficial noise or patterns (e.g., grammar changes or low-level visual features), whereas the update from edit's invariant semantics is generalizable. The knowledge space approximates the underlying semantic manifold of the new knowledge, leveraging low-rank representations of the model's internal states as reliable signals of the manifold's structure (Figure 4b).

In this paper, we conduct extensive experiments to evaluate the editing performance of ELoRA on an editing visual question answer task involving simple triplet knowledge and a visual entity editing task involving complex free-form unstructured knowledge. We leverage rigorous LLM-as-a-Judge (Zheng et al., 2023) evaluation, which reflects real-world knowledge update capability (Yang et al., 2025), along with previous token-level evaluation. As shown in Figure 1a, **ELoRA achieves balanced superiority across reliability, generality, and locality in real-world evaluation**, significantly outperforming existing LoRA-based methods. ELoRA boosts average accuracy across all metrics from 69.7% to 83.9% under LLM-as-a-Judge evaluation, which highlights the effectiveness of our subspace decomposition design in both localized and generalizable knowledge editing over LoRA.

## 2 RELATED WORK

**Multimodal Knowledge Editing.** Knowledge editing methods are typically categorized into two paradigms: external knowledge retrieval (Zheng et al.; Pan et al., 2024) and intrinsic parameter modification (Meng et al., 2022; Meng et al.; Deng et al., 2025; Fang et al., 2025; Yu et al., 2024; Wang et al., 2024; Li et al., 2025). Our work focuses on the latter, particularly on learning low-rank matrices to incorporate new knowledge (Yu et al., 2024). AlphaEdit (Fang et al., 2025) utilizes null-space projection within the locate-then-edit paradigm to handle knowledge triplets, whereas UnKE (Deng et al., 2025) employs direct parameter optimization for more common unstructured knowledge edits. While these techniques have demonstrated promising results in LLMs, their adaptation to multimodal settings remains underexplored. UniKE (Pan et al., 2024) disentangles

the knowledge representations into the semantic and truthfulness spaces, and Multi-MELO (Chen et al., 2025) dynamically activates LoRA blocks that encode the related knowledge. However, a key limitation of these methods persists: the trade-off between the generality and locality of edits. This challenge is intensified in multimodal contexts, where models must generalize across diverse perceptual variations and precisely localize edits to specific cross-modal associations. Our work tackles the issue by employing decomposed subspaces in low-rank updates, which separately ensure multimodal invariances in localized and generalizable knowledge.

**Parameter-Efficient Fine-Tuning.** PEFT methods reduce the number of trainable parameters by introducing lightweight modules into large models (Han et al., 2024). LoRA (Hu et al.) is the representative one that achieves efficient fine-tuning by reparameterizing weight updates into low-rank matrices. Some variants like CorDA (Yang et al., 2024) and LoRA-Null (Tang et al., 2025) explore context-oriented decomposition and null-space projection to balance downstream task adaptability with resistance to catastrophic forgetting. However, balancing locality and generality in knowledge editing remains challenging, as the objective is to precisely update general knowledge with few samples, rather than adapting to downstream tasks. To this end, we propose a subspace decomposition approach that improves LoRA for effective knowledge editing.

## 3 METHODOLOGY

In this section, we introduce model editing problems and the preliminaries of LoRA in Section 3.1. We provide an overview of our decomposed editing space for knowledge editing in Section 3.2, which is composed of the null space (Sections 3.3) and the low-rank knowledge space (Section 3.4).

### 3.1 PRELIMINARIES

**Problem definition.** We consider a base model $f : \mathcal{X} \to \mathcal{Y}$ and a specific edit example $(x_e, y_e)$, where the model's current prediction $f(x_e) \neq y_e$ is incorrect. The objective of model editing is to apply an update procedure based on this example to transform the base model $f$ into a revised version $f'$ (i.e., $f \to f'$). A successful editing procedure should ensure that the resulting model $f'$ satisfies the following properties:

- **Property 1 - Reliability**: The edited model $f'$ must provide the target output for the edit input:
$$f'(x_e) = y_e$$

- **Property 2 - Generality**: The edit should generalize to a set of semantically equivalent inputs $S_{x_e} = \{x_j | y_{x_j} = y_e\}$. For any input in this set, the model should produce the target output:
$$\forall x_j \in S_{x_e}, \quad f'(x_j) = y_e$$

- **Property 3 - Locality**: The edit should not alter the model's predictions on unrelated inputs. For the set of all other inputs $O_{x_e} = \mathcal{X} \setminus S_{x_e}$, the model's behavior must remain unchanged:
$$\forall x_j \in O_{x_e}, \quad f'(x_j) = f(x_j)$$

In model editing, achieving reliability, generality, and locality simultaneously faces a trilemma, and this challenge stems from catastrophic forgetting (French, 1999) (i.e., poor locality) as well as the inherent complexities of the learning dynamics (Ren & Sutherland, 2025) (i.e., the fine-tuning imbalance between reliability and generality).

**Low-Rank adaptation (LoRA).** Instead of updating the full weight matrix, LoRA (Hu et al.) freezes the pre-trained weights $\mathbf{W} \in \mathbb{R}^{d_{\text{out}} \times d_{\text{in}}}$ and learns a low-rank decomposition of the weight update as $\Delta \mathbf{W} = \mathbf{BA}$ for each layer, where $\mathbf{B} \in \mathbb{R}^{d_{\text{out}} \times r}$ and $\mathbf{A} \in \mathbb{R}^{r \times d_{\text{in}}}$ are learnable low-rank matrices, with the rank $r \ll \min(d_{\text{out}}, d_{\text{in}})$. Thus, the fine-tuned weight matrix $\mathbf{W}^*$ of the layer is: $\mathbf{W}^* = \mathbf{W} + \mathbf{BA}$. LoRA significantly reduces the number of trainable parameters from $d_{\text{out}} \times d_{\text{in}}$ to $r \times (d_{\text{out}} + d_{\text{in}})$. Typically, $\mathbf{A}$ is initialized with Kaiming initialization (He et al., 2015), and $\mathbf{B}$ is initialized to zero, ensuring $\Delta \mathbf{W} = \mathbf{0}$ at the start of training to maintain the pre-trained knowledge.

However, directly applying LoRA to model editing poses challenges, since editing typically relies on only one (or a few) samples rather than sufficient fine-tuning data, making fine-tuning prone to overfitting sample-specific noise and less effective at capturing the intended knowledge change.

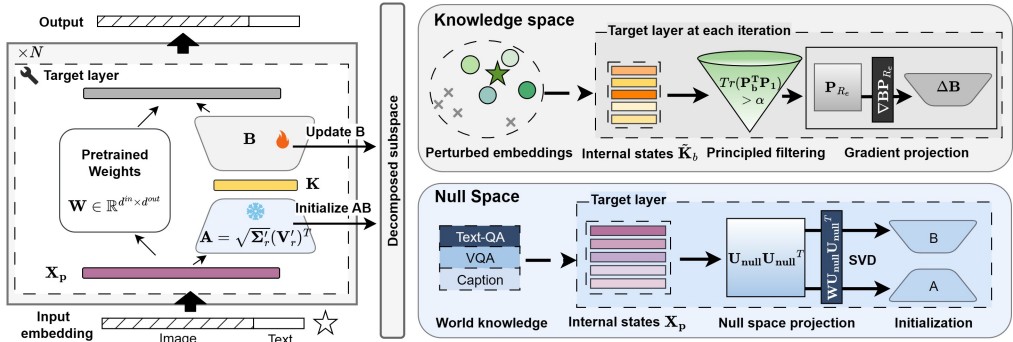

Figure 2: Overview of ELoRA. Left: ELoRA updates $\mathbf{B}$ by projecting it onto the knowledge space, whereas $\mathbf{A}$ is initialized in the null space and kept frozen. Right: Construction of the knowledge space and the null space. Specifically, we incorporate textual and visual question-answer pairs (TQA and VQA) along with image caption pre-trained data to encode world knowledge.

## 3.2 DECOMPOSED EDITING SPACE

To tackle the above trilemma in Section 3.1, we decompose the model editing space into two complementary subspaces: *the null space* (Wang et al., 2021), which preserves locality by preventing interference with unrelated knowledge, and *the knowledge space*, which is responsible for enhancing reliability and generality of relevant knowledge, as shown in Figure 2.

Specifically, given input $\mathbf{X} \in \mathbb{R}^{N \times d_{\text{in}}}$ and pre-trained weight $\mathbf{W} \in \mathbb{R}^{d_{\text{out}} \times d_{\text{in}}}$ for the target edited layer, the layer's output is computed as $\mathbf{X}\mathbf{W}^{\mathbf{T}}$. N denotes the sequence length, $d_{in}$ and $d_{out}$ are dimensions of the weight matrix. Considering a typical editing objective in LMMs, the weight update $\Delta\mathbf{W} \in \mathbb{R}^{d_{\text{out}} \times d_{\text{in}}}$ is optimized by editing samples to **achieve reliability**:

$$\Delta\mathbf{W} = -\eta\nabla_{\mathbf{W}}\mathcal{L}(x_e, y_{x_e}) \tag{1}$$

where $\eta$ denotes the learning rate, and $\mathcal{L}$ is commonly the negative log-likelihood loss function.

**Null space (preserving locality).** Given two matrices $\mathbf{A}$ and $\mathbf{B}$, $\mathbf{B}$ is in the null space of $\mathbf{A}$ if and only if $\mathbf{BA} = 0$ (Wang et al., 2021). Specifically in model editing, given unrelated inputs $\mathbf{X}_p \in R^{N \times d_{in}}$ for a specific layer, preserving locality requires the layer's output to remain unchanged:

$$\mathbf{X_p}(\mathbf{W} + \Delta\mathbf{W})^T = \mathbf{X_p}\mathbf{W}^{\mathbf{T}} \quad \Longrightarrow \quad \mathbf{X_p}\Delta\mathbf{W}^{\mathbf{T}} = \mathbf{0} \tag{2}$$

Eq.2 implies that $\Delta W^T$ must lie in the null space of $X_p$, denoted as $\mathcal{N}_p = \text{Null}(\mathbf{X_p})$. The locality constraint is formally defined as requiring the row space[1] of $\Delta W$ to be a subspace of $N_p$:

$$\text{Row}(\Delta\mathbf{W}) \subseteq \mathcal{N}_p \tag{3}$$

**Knowledge space (enhancing generality).** Typical gradient descent of $\Delta\mathbf{W}$ in Eq.1 pulls $\Delta\mathbf{W}$ along the directions of the row space of one (or a batch of) editing input $\mathbf{X_e} \in \mathbb{R}^{N_e \times d_{in}}$. To accommodate knowledge changes, we further refine the space $R_e = \text{Row}(\mathbf{X_e})$ by incorporating generalizable yet concentrated editing knowledge in the form of abstract semantics (Section 3.4). Thus, effective knowledge updates require to constrain the row space of $\Delta\mathbf{W}$ to be a subspace of $R_e$. To strike a balance between enhancing reliability/generality and preserving locality, the weight update $\Delta\mathbf{W}$ should satisfy both constraints:

$$\text{Row}(\Delta\mathbf{W}) \subseteq N_p \cap \mathcal{R}_e \tag{4}$$

**Decomposed editing space with LoRA.** We leverage LoRA to update the weight $\Delta W$ and decompose the editing space by explicitly targeting the subspace $N_p \cap \mathcal{R}_e$ using $\mathbf{A}$ and $\mathbf{B}$ matrices.

First, the null space $\mathcal{N}_p$, as conditioned in Eq.2, requires that $\mathbf{A}$ and $\mathbf{B}$ satisfy:

$$\mathbf{X_p}(\mathbf{BA})^T = 0 \tag{5}$$

---

[1]The row space of a matrix $\mathbf{X} \in \mathbb{R}^{m \times n}$ is the subspace of $\mathbb{R}^n$ spanned by its row vectors, denoted as Row($\mathbf{X}$). It represents all possible linear combinations of the rows of $\mathbf{X}$.

Inspired by LoRA's matrices playing asymmetric roles, and that fine-tuning only $\mathbf{B}$ can achieve performance comparable to fine-tuning both $\mathbf{A}$ and $\mathbf{B}$ (Yang et al., 2024). We freeze the $\mathbf{A}$ during the editing to serve the role of preservation, ensuring it remains in the null space $\mathcal{N}_p$ (Section 3.3):

$$\mathbf{X_p}\mathbf{A}^T = 0 \tag{6}$$

Second, during the editing, the matrix $\mathbf{B}$ is fine-tuned and projected onto the knowledge space $\mathcal{R}_e$, which is defined by the projection matrix $\mathbf{P}_{R_e}$, at each iteration. Since $\mathbf{A}$ is frozen, the weight update is given by:

$$\Delta\mathbf{W} = (\Delta\mathbf{B}\mathbf{P}_{R_e})\mathbf{A} \tag{7}$$

Eq.7 ensures that the weight changes remain aligned with the relevant knowledge (Section 3.4).

### 3.3 INITIALIZATION OF NULL SPACE

To ensure locality, we adapt previous null-space projection techniques (Tang et al., 2025; Fang et al., 2025) in multimodal context. Considering that the null space constructed solely from text is blind to cross-modal knowledge in LMMs, *we construct a multimodal-aware null space from representative text-image pairs*. We note that initializing $\mathbf{A}$ in the multimodal-aware null space and freezing it serves as a critical guarantee for the edits within the proposed knowledge space (Section 3.4). The procedure to construct the multimodal-aware null space is as follows.

First, we collect a few samples $x_p$, consisting of text-image pairs from VQA and image caption datasets, to represent the pre-trained knowledge and obtain the input activations of these samples at the target layer, denoted as $\mathbf{X_p} \in \mathbb{R}^{N_p \times d_{in}}$. Then we compute the covariance matrix $\mathbf{C} = \mathbf{X_p}^T\mathbf{X_p} \in \mathbb{R}^{d_{in} \times d_{in}}$, as shown in Figure 2.

Second, we apply Singular Value Decomposition (SVD) to C, yielding $\mathbf{C} = \mathbf{U}\mathbf{\Sigma}\mathbf{V}^T$, and extract the columns of $\mathbf{U}$ corresponding to small singular values in $\mathbf{\Sigma}$ to form the null space $\mathbf{U}_{\text{null}}$ of the preserved samples, which spans a subspace with minimal variance.

Third, we project the weight update $\Delta\mathbf{W}$ of the target edited layer onto the null space. The projected weight matrix $\Delta\mathbf{W}_{\text{proj}}$ is given by $\Delta\mathbf{W}\mathbf{U}_{\text{null}}\mathbf{U}_{\text{null}}^T$.

Finally, we perform SVD on $\Delta\mathbf{W}_{\text{proj}}$, yielding $\Delta\mathbf{W}_{\text{proj}} = \mathbf{U}'\mathbf{\Sigma}'(\mathbf{V}')^T$, and use the top-$r$ singular values and vectors $(\mathbf{\Sigma}'_r, \mathbf{U}'_r, \mathbf{V}'_r)$ to initialize the LoRA matrices as:

$$\mathbf{B} = \mathbf{U_r}'\sqrt{\mathbf{\Sigma}'_r}, \mathbf{A} = \sqrt{\mathbf{\Sigma}'_r}(\mathbf{V}'_r)^T \tag{8}$$

Thus, Eq.8 satisfies the condition in Eq.5. The pre-trained weight matrix $\mathbf{W}$ is further replaced by $\mathbf{W} - \mathbf{B}\mathbf{A}$ to avoid modifying the pre-trained weights at the initial fine-tuning.

### 3.4 MODEL EDITING IN KNOWLEDGE SPACE

**Identifying the invariant knowledge manifold.** The conventional editing methods directly optimize parameters from a single edit sample $(x_e, y_{x_e})$, which tend to overfit to irrelevant features, thereby restricting generality. To address this, we hypothesize that *the target knowledge is encoded within a low-dimensional, invariant manifold in the model's internal states*. This assumption directly builds upon the linear representation hypothesis (Park et al., 2024), as evidenced by ROME Meng et al. (2022), which successfully edits factual associations using rank-one updates, suggesting that simple atomic facts are localized in a single editing direction. However, multimodal knowledge is more entangled, making rank-one updates insufficient. Recent interpretability studies (Modell et al., 2025) further indicate that many features in large models form *continuous, nonlinear manifolds* rather than isolated directions. Thus, our objective is to first identify the underlying manifold and then confine the parameter update $\Delta\mathbf{B}$ within it, as shown in Figure 2.

We approximate the targeted manifold with a linear knowledge subspace to make the problem trivial. The knowledge subspace is derived from the internal states, i.e., input activation of $\mathbf{B}$, with perturbed multimodal embeddings: applying random masking to visual embeddings and injecting Gaussian

noise into textual embeddings (as detailed in Appendix C.2), creating a distribution of inputs $\{x_b\}$ around the original $x_0$. For each input, we extract the internal states $\tilde{K}_b \in \mathbb{R}^{N \times r}$ for the LoRA matrix $\mathbf{B}$, as visualized in Figure 4b. The covariance matrix of these activations, $\mathbf{C}_b = \tilde{K}_b \tilde{K}_b^T$ characterizes the second-moment geometry of the model's internal states to each perturbation.

However, as not all perturbations are semantically faithful, we introduce a *principled filtering mechanism* to discard unreliable covariance matrices that could corrupt the edit to ensure invariant semantics. This is achieved by computing the principal components $P_b$ of each covariance matrix $C_b$ and assessing their alignment with those of the original sample $P_1$ via a similarity metric:

$$s_b = \mathrm{Tr}(\mathbf{P}_b^T \mathbf{P}_1) \tag{9}$$

where $\mathrm{Tr}(\cdot)$ denotes the trace of a matrix. If the similarity $s_b$ of $\mathbf{P}_b$ falls below a predefined threshold $\theta$, indicating that the model's states change drastically from the original states, the corresponding $\mathbf{C}_b$ is discarded, *as its perturbation may compromise the core semantics of the knowledge*. The set of retained covariance matrices is denoted as $S_C$.

We average $S_C$ to represent the knowledge-related covariance matrix $\mathbf{C}_{\mathrm{avg}}$ and perform SVD on it: $\mathbf{C}_{\mathrm{avg}} = \mathbf{U}\mathbf{\Sigma}\mathbf{V}^T$. The top-$k$ left singular vectors are selected to form $U_{\mathrm{know}} \in \mathbb{R}^{r \times k}$, which spans the knowledge subspace. The projection matrix $\mathbf{P}_{R_e}$ onto this subspace is constructed as:

$$\mathbf{P}_{R_e} = \mathbf{U}_{\mathrm{know}}\mathbf{U}_{\mathrm{know}}^T \in \mathbb{R}^{r \times r} \tag{10}$$

**Gradient projection in knowledge space.** We project the gradient of $\mathbf{B}$ during editing onto the knowledge subspace, thereby constraining the parameter updates. Specifically, during each editing iteration, we project raw gradient $\nabla \mathbf{B}$ using $\mathbf{P}_{R_e}$ as Eq.11, where $\eta$ is the learning rate.

$$\nabla \mathbf{B}_{\mathrm{proj}} = \nabla \mathbf{B} \mathbf{P}_{R_e}, \mathbf{B} \leftarrow \mathbf{B} - \eta \nabla \mathbf{B}_{\mathrm{proj}} \tag{11}$$

A detailed algorithmic description of `ELoRA` can be found in Appendix C.

## 4 EXPERIMENTS

In this section, we conduct experiments to address the following research questions:

- **RQ1**: How does `ELoRA` perform in knowledge updates and knowledge preservation compared with baseline methods?
- **RQ2**: Do the covariance matrices $C_b$ from varied perturbed internal states inform the steering of parameter updates in knowledge space?
- **RQ3**: What is the individual and combined contribution of the proposed components (i.e., null space and knowledge space) to the performance of multimodal knowledge editing?

### 4.1 EXPERIMENTAL SETUP

We briefly introduce the datasets, baseline methods, and evaluation metrics in this section. Detailed experimental settings are provided in Appendix G.

**Datasets & Base LMMs** We conduct experiments on Editing Visual Question Answering (*E-VQA*) from the MMEdit benchmark (Cheng et al., 2023) and the visual entity editing task from MMKE-Bench (Du et al.). Specifically, E-VQA provides simple triplet-based knowledge representations (i.e., subject, relation, object), focusing on direct factual querying and updating. MMKE-Bench (entity) provides free-form real-world multimodal knowledge, utilizing counterfactual editing to construct challenging scenarios that require nuanced reasoning over altered facts. We evaluate knowledge editing methods on LLaVA-v1.5-7B (Liu et al., 2023), Qwen2.5-VL-7B (Team, 2025), and Phi-4-multimodal (Abouelenin et al., 2025). To align with real-world scenarios, we incorporate task-specific instructions in the prompts and use autoregressive decoding for output generation, as detailed in Appendix G.2.

**Baselines.** We compare ELoRA with a variety of editing baselines, including *intrinsic knowledge editing*: AlphaEdit (Fang et al., 2025), UnKE (Deng et al., 2025), *multimodal knowledge editing*: UniKE (Pan et al., 2024), Multi-MELO (Chen et al., 2025), and *parameter-efficient fine-tuning*: LoRA (Hu et al.), AdaLoRA (Zhang et al.), RoseLoRA (Wang et al., 2024), CorDA (Yang et al., 2024), LoRA-Null (Tang et al., 2025). Detailed settings of baselines are provided in Appendix G.3.

**Metrics.** We evaluate all editing methods using three standard metrics (Zhang et al., 2024): (1) Reliability measures whether the updated knowledge is successfully incorporated. (2) Generality assesses the persistence of the edit across various rephrased textual prompts and images. (3) Locality verifies that knowledge unrelated to the edit remains intact. For generality and locality, we consider both textual and visual modalities, i.e., respectively, T-Generality and M-Generality for generality, and T-Locality and M-Locality for Locality.

**LLM-as-a-Judge.** Consistent with prior work (Pan et al., 2024; Chen et al., 2025), we report token-level accuracy. However, conventional teacher forcing in token-level accuracy by feeding ground truth tokens prevents error propagation, potentially limiting the metrics' reflection on real-world performance (Yang et al., 2025). Thus, we employ a more rigorous evaluation criterion, leveraging the LLM-as-a-Judge protocol (Zheng et al., 2023). Specifically, we employ Qwen2.5-Turbo (Qwen et al., 2025) to compare each post-edit generated response with the editing target for reliability and generality, or with the model's pre-edit output for locality, yielding a binary verdict (i.e., correct/incorrect) to indicate the real-world editing performance, as detailed in Appendix G.2.

## 4.2 PERFORMANCE ON KNOWLEDGE UPDATE AND PRESERVATION (RQ1)

Table 1: Comparison of ELoRA with existing methods on two benchmarks: E-VQA (Cheng et al., 2023) and MMKE-Bench (Du et al.). We report the following evaluation metrics: Reliability, T-Generality (T-Gen), M-Generality (M-Gen), T-Locality (T-Loc), and M-Locality (M-Loc). "Real." refers to evaluations under the LLM-as-a-Judge protocol, which is our main concern. "Edit." denotes token-level testing accuracy. "Real. Avg" is the average score across all metrics evaluated under LLM-as-a-Judge (Zheng et al., 2023).

| Method | Reliability | | T-Gen | | M-Gen | | T-Loc | | M-Loc | | Real. Avg |
|---|---|---|---|---|---|---|---|---|---|---|---|
| | Real. | Edit. | Real. | Edit. | Real. | Edit. | Real. | Edit. | Real. | Edit. | |
| **E-VQA** | | | | | | | | | | | |
| **Intrinsic & multimodal editing methods** | | | | | | | | | | | |
| ROME (Meng et al., 2022) | 81.27 | 90.06 | 59.53 | 65.28 | 59.29 | 71.74 | 85.19 | 98.04 | 91.35 | 95.89 | 75.33 |
| MEMIT (Meng et al.) | 71.09 | 90.24 | 69.42 | 78.31 | 60.25 | 78.55 | 77.50 | 94.97 | 87.53 | 87.58 | 73.15 |
| AlphaEdit (Fang et al., 2025) | 47.01 | 38.63 | 47.01 | 31.17 | 29.67 | 30.44 | 89.54 | 98.02 | **95.89** | 97.83 | 61.82 |
| UnKE (Deng et al., 2025) | 93.41 | 90.18 | 75.01 | 67.77 | **71.33** | 72.48 | 88.82 | 97.29 | 93.74 | 89.82 | **84.46** |
| UniKE (Pan et al., 2024) | 65.79 | 75.24 | 56.00 | 63.99 | 53.46 | 63.59 | 9.99 | 68.60 | 43.72 | 75.72 | 45.79 |
| Multi-MELO (Chen et al., 2025) | 51.12 | 82.39 | 42.52 | 70.62 | 41.14 | 66.49 | 91.40 | 99.35 | 94.08 | **98.11** | 64.05 |
| **Parameter-efficient fine-tuning methods** | | | | | | | | | | | |
| LoRA (Hu et al.) | 62.54 | **100.0** | 66.89 | **98.61** | 54.52 | 93.69 | 68.99 | 48.21 | 35.77 | 38.51 | 57.74 |
| AdaLoRA (Zhang et al.) | 89.11 | 98.52 | 75.49 | 80.68 | 60.25 | 76.65 | 85.09 | 96.24 | 77.35 | 59.44 | 77.46 |
| RoseLoRA (Wang et al., 2024) | 86.24 | 94.27 | 62.40 | 85.15 | 48.73 | 65.36 | 87.82 | 97.80 | 83.95 | 70.35 | 73.83 |
| CorDA (Yang et al., 2024) | 74.30 | **100.0** | 77.07 | 95.65 | 62.35 | 91.00 | 65.31 | 81.02 | 58.15 | 39.53 | 67.44 |
| LoRA-Null (Tang et al., 2025) | 92.45 | 99.10 | 79.60 | 83.98 | 58.19 | 74.56 | 60.54 | 87.53 | 69.18 | 43.94 | 71.99 |
| ELoRA(Ours) | **93.74** | **100.0** | 80.46 | 80.62 | 63.12 | 77.57 | **97.09** | 96.86 | 84.90 | 64.69 | 83.86 |
| **MMKE-Bench** | | | | | | | | | | | |
| **Intrinsic & multimodal editing methods** | | | | | | | | | | | |
| AlphaEdit (Fang et al., 2025) | 1.26 | 59.61 | 0.21 | 58.8 | 0.63 | 59.42 | **90.47** | **98.27** | **92.67** | 99.21 | 37.05 |
| UnKE (Deng et al., 2025) | 67.64 | 99.14 | 27.75 | 94.42 | 60.42 | 98.77 | 87.43 | 97.81 | 92.57 | **99.37** | 67.16 |
| UniKE (Pan et al., 2024) | 11.94 | 87.04 | 8.48 | 86.06 | 8.80 | 86.15 | 8.59 | 71.36 | 34.45 | 78.90 | 14.45 |
| **Parameter-efficient fine-tuning methods** | | | | | | | | | | | |
| LoRA (Hu et al.) | 75.71 | 99.23 | 71.94 | 98.77 | 71.94 | 99.20 | 37.07 | 84.56 | 44.71 | 87.43 | 60.27 |
| AdaLoRA (Zhang et al.) | **86.39** | **100.0** | **85.65** | **99.71** | **87.02** | **99.99** | 49.11 | 92.11 | 53.40 | 91.46 | 72.31 |
| RoseLoRA (Wang et al., 2024) | 81.75 | 99.85 | 43.25 | 98.61 | 63.25 | 99.60 | 56.69 | 95.12 | 81.78 | 96.94 | 65.34 |
| CorDA (Yang et al., 2024) | 75.08 | 99.21 | 74.14 | 98.75 | 71.73 | 99.19 | 34.45 | 83.02 | 48.48 | 86.84 | 60.78 |
| LoRA-Null (Tang et al., 2025) | 81.88 | **100.0** | 85.97 | 99.38 | 80.10 | 99.96 | 19.48 | 57.12 | 43.35 | 77.31 | 62.16 |
| ELoRA (Ours) | 85.97 | **100.0** | 85.55 | 99.55 | 84.50 | 99.98 | 89.95 | 94.12 | 89.01 | 96.39 | **87.00** |

To evaluate the performance of various editing and PEFT methods in preserving and updating knowledge, we conduct experiments on two multimodal knowledge editing benchmarks: E-VQA (Cheng

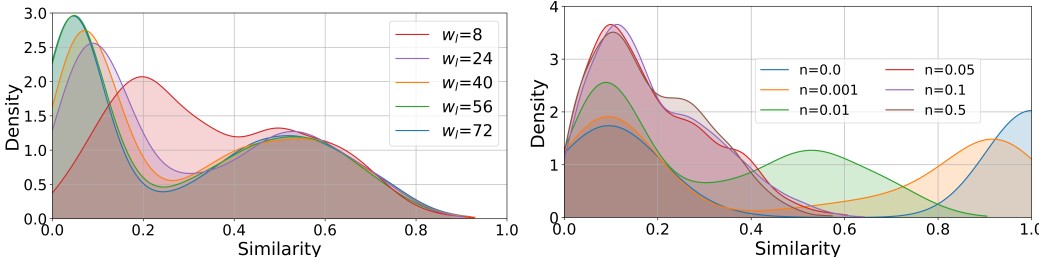

(a) Similarity distribution under different window length.(b) Similarity distribution under different noise variance.

Figure 3: The impact of perturbed editing inputs on the knowledge representation. The x-axis denotes the similarity score $s_b$, and the y-axis indicates the kernel density estimate.

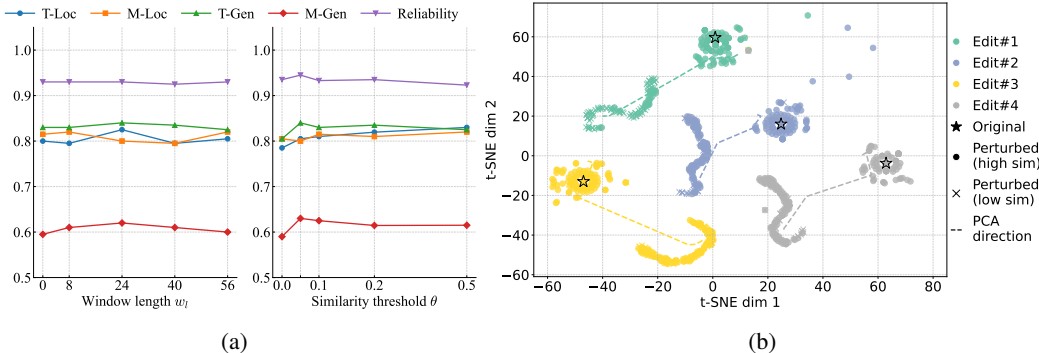

(a)                                    (b)

Figure 4: (a) Real-world performance (y-axis) under varying window length and similarity threshold. (b) t-SNE visualization of the activation space for four edit instances, distinguished by color. For each edit, the star ($\star$) is the original sample's internal state, while perturbed activations are represented as circles ($\bullet$) and crosses ($\times$), denoting high and low similarity with the original sample, respectively. The "PCA direction" dashed line indicates the first principal component of the activations, defining the primary dimension of the knowledge subspace.

et al., 2023) and MMKE-Bench (Du et al.). The results are shown in Table1 and Appendix A. Each edit involves updating knowledge based on a single editing sample in a single layer, e.g., the 7th layer. More ablations of layer-wise editing are in Appendix B. We make two key observations:

- **Obs1**: `ELoRA` **achieves balanced superiority across reliability, generality, and locality** when editing triplet-based knowledge in E-VQA under LLM-as-a-Judge evaluation. `ELoRA` surpasses other PEFT methods by an average of 14.17% and is only 0.6% behind UNKE because UNKE directly modifies the original model parameters with the editing objective. Particularly, `ELoRA` improves reliability by an average of 26.12% over its base method, LoRA, with an average 11.08% gain in generality and 38.62% in locality.
- **Obs2**: `ELoRA` leverages the strengths of PEFT approaches to update free-form knowledge in MMKE-Bench, while **effectively preserving locality across both textual and visual modalities constrained in the null space.** Specifically, UnKE (Deng et al., 2025) exhibits poor performance in terms of reliability and generality. Compared with editing methods, PEFT methods typically achieve high performance in reliability and generality, but suffer from poor locality. For example, PEFT methods record an average of 39.36% in T-Locality and 54.34% in M-Locality under LLM-as-a-Judge evaluation, which are both lower than the corresponding 88.95% and 92.62% of editing methods (excluding UniKE). `ELoRA` improves T-Locality and M-Locality by 50.59% and 34.67% through isolating null space constraints from edit updates.

### 4.3 ANALYSIS OF KNOWLEDGE SPACE (RQ2)

**Visualization of knowledge space construction** Figure 4b provides an intuitive validation of our knowledge space construction. Notably, for each edit instance, the activations form two distinct sub-clusters, corresponding to the model's internal states of text and vision perturbations, respectively. The key insight is that our method's PCA direction (the dashed line) successfully identifies a single, shared

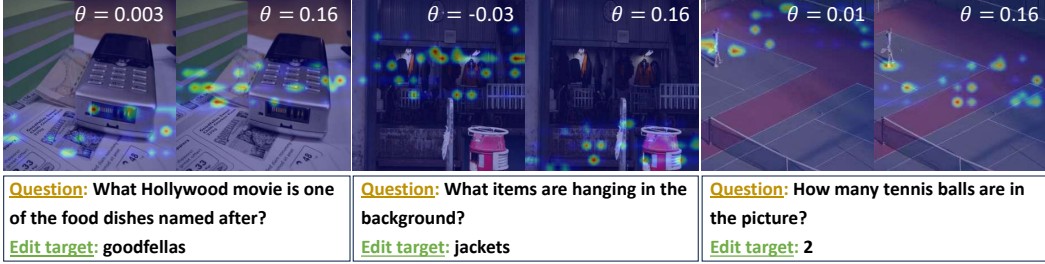

Figure 5: Visualization of the spatial impact of visual perturbations. Gradient-weighted mapping is used to project embedding deviations back into the image space. The heatmaps highlight regions where perturbed embeddings deviate the most from the originals. Samples with low similarity scores (e.g., $\theta \approx 0$) show substantial disruption of key visual subjects (e.g., the jackets or tennis ball), indicating reduced semantic consistency.

semantic axis that spans both of these sub-clusters, which demonstrates the ability of knowledge space to capture the core, abstract semantics of an edit.

**Analysis of perturbation hyperparameters**   We analyze the impact of perturbed editing inputs on the knowledge representation, as shown in Figure 3. We separately investigate the effects of two hyperparameters mentioned in §3.4: the window length $w_l$ for visual knowledge representation and the noise variance $\sigma$ for textual knowledge representation. We observe that adjusting the degree of multimodal perturbation affects the contribution to the knowledge space in terms of similarity. The effects are twofold. First, *as the window length enlarges, lower similarity becomes dominant* (e.g., an upper-left shift in Figure 3a). Second, *as the noise variance increases, the similarity does not become lower like perturbing visual token embeddings*, but the proportion of low similarity exhibits higher (e.g., an upper shift in Figure 3b). The observed variations in contribution underscore the need for separate perturbation of visual and textual modalities to better represent multimodal generality. Introducing covariance matrices with lower similarity into the knowledge space enriches the variance of the gradient projection in Eq.11.

**Interpretability analysis of multimodal perturbations**   To gain insight into how knowledge space isolates invariant multimodal knowledge from disruptive noise, we visualize the impact of visual masking on the visual embeddings in Figure 5 and quantify the semantic variation of perturbed texts in Appendix F. For visual perturbations, we map the masked visual embeddings $\mathbf{Z}$ (denoted in Appendix C.2) back to the pixel space using a gradient-based attribution method inspired by Grad-CAM (Selvaraju et al., 2017). Specifically, we calculate the gradients of the Euclidean distance ($\ell_2$ distance) between the perturbed embeddings $\{\mathbf{Z}_b\}$ and the original embedding $\mathbf{Z}_0$ with respect to the $24 \times 24$ visual patches. These patch-level gradients are then interpolated to the original image resolution ($336 \times 336$) to generate the attribution heatmaps.

As illustrated in Figure 5, we observe a correspondence between the similarity score $\theta$ and the perturbed regions. Instances with low similarity scores (e.g., $\theta \approx 0$) frequently exhibit high activation deviations concentrated on semantically key subjects (e.g., the jackets required to answer the question). This suggests that random masking in these specific cases likely disrupted the core visual features needed for the editing task. Therefore, filtering out these deviated samples helps ensure that the constructed knowledge space remains aligned with the intended semantic content.

## 4.4 ABLATION STUDY (RQ3)

**Effect of null space and knowledge space.**   To investigate the effectiveness of the proposed null space and knowledge space, we conduct a detailed ablation study. Table2 reports the results of different module combinations of ELoRA on E-VQA. Three observations are as follows:

- **Obs3**: **Null space significantly boosts reliability and locality.** Compared to $M_0$, $M_1$ and $M_2$ initialize matrix $\mathbf{A}$ (frozen) in multimodal knowledge's null space, improving reliability from 62.54% to 86.53%, 87.29%, and average locality from 52.38% to 85.28%, 86.13%, but slightly degrades visual generality, which is due to the constrained updated space beyond the null space.

- **Obs4**: **Fine-tuning matrix A during editing improves generality but degrades locality drastically.** Compared with $M_2$, the average locality of $M_3$ drops from 86.13% to 61.35%. The degradation in locality is primarily due to the fact that, by unfreezing **A** during editing, its parameters are updated and no longer reside in the null space of the preserved knowledge.
- **Obs5**: **Knowledge Space effectively mitigates the degradation in generality caused by the null space.** By projecting the gradient updates onto the knowledge space constructed by visual and textual perturbations, the model not only recovers generality (even better) but also achieves the best overall results across all metrics. Compared with $M_2$, $M_5$ achieves 8.81% overall increase.

Table 2: Ablation study of ELoRA with different module combinations under LLM-as-a-Judge evaluation across three types of metrics. "KnowSpace" is short for knowledge space. "VL Init.", "Language Init." and "Vision Init." refer to initializing the null or knowledge space by incorporating both modalities, text alone, or vision alone, respectively.

| ID | Method Description | Reliability | T-Gen | M-Gen | T-Loc | M-Loc |
|----|--------------------|-------------|-------|-------|-------|-------|
| $M_0$ | Base LoRA | 62.54 | 66.89 | 54.52 | 68.99 | 35.77 |
| $M_1$ | $M_0$ + Null Space (Language Init.) | 86.53 | 67.70 | 46.34 | 88.29 | 82.27 |
| $M_2$ | $M_0$ + Null Space (VL Init.) | 87.29 | 68.18 | 47.54 | 87.63 | 84.62 |
| $M_3$ | $M_2$ + **A** unfrozen | 78.64 | 78.50 | **64.74** | 64.12 | 58.58 |
| $M_4$ | $M_2$ + KnowSpace (Language Init.) | 93.02 | **83.05** | 59.50 | 80.34 | 81.50 |
| $M_5$ | $M_2$ + KnowSpace (VL Init.) | **93.74** | 80.46 | 63.12 | **97.09** | **84.90** |

Additionally, we present results across different similarity thresholds $\theta$ and window lengths $w_l$ in knowledge space as Figure 4a. We find that as the similarity threshold increases, locality exhibits an increasing trend, while generality first increases and then declines, which indicates that *introducing much more perturbed semantics in the knowledge space will compromise both locality and generality*. Meanwhile, when the similarity threshold is fixed (e.g., 0.05), varying the window length has minimal impact on overall performance.

## 5 CONCLUSION

This paper presents a novel solution to the core challenges in multimodal knowledge editing within the parameter-efficient fine-tuning paradigm. ELoRA introduces decomposed subspaces for projecting matrices in the original LoRA, enabling the simultaneous preservation of pre-trained knowledge and acquisition of generalizable knowledge. The resulting method achieves balanced superiority over previous SOTA, performing consistently well across both triplet-based and free-form knowledge editing tasks, as evaluated by rigorous LLM-as-a-Judge protocols. Our work provides a promising direction to achieve reliable and controllable knowledge editing using PEFT methods. The statement of LLM usage is put in Appendix I.

## ETHICS STATEMENT

The development of reliable knowledge editing techniques for multimodal models, such as our proposed `ELoRA`, is crucial for maintaining the accuracy and safety of these models over time. We recognize that any method capable of altering a model's internal knowledge also carries the potential for misuse, including the insertion of biased or factually incorrect text-image associations. Despite these risks, the fundamental motivation for this field of research is constructive: to provide an efficient and targeted mechanism for correcting, updating, and refining the knowledge within large multimodal models. It is with this positive goal in mind that we present our work, and we strongly encourage the community to continue developing these powerful techniques with responsibility and foresight.

## REPRODUCIBILITY STATEMENT

To ensure the reproducibility of our findings, detailed implementation instructions for `ELoRA` can be found in Appendix C.1. Comprehensive implementation details, including the experimental setup and hyperparameters, are provided in Appendix G. To enable complete reproduction of our work, we will release our code after the review process is completed.

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

## A    ADDITIONAL RESULTS ON LMMS

To further demonstrate the generality of **ELoRA**, we conduct additional experiments on two state-of-the-art LMMs: **Qwen2.5-VL-7B** and **Phi-4-multimodal**. The results show that **ELoRA consistently outperforms all PEFT-based methods on average under real-world evaluation**. All metrics reported in Table 3 and Table 4 are real-world metrics, evaluated along five dimensions: Reliability, T-Generality (T-Gen), M-Generality (M-Gen), T-Locality (T-Loc), and M-Locality (M-Loc).

### A.1    RESULTS ON QWEN2.5-VL-7B

| Methods | Reliability | T-Gen | M-Gen | T-Loc | M-Loc | Real. Avg |
|---------|------------|-------|-------|-------|-------|-----------|
| LoRA | **99.28** | 81.99 | 68.56 | 68.42 | 87.91 | 81.23 |
| AdaLoRA | 99.24 | 81.94 | 68.47 | 67.18 | 88.01 | 80.97 |
| RoseLoRA | 66.10 | 43.64 | 29.24 | 69.83 | 92.97 | 60.36 |
| LoRA-Null | 99.14 | 56.86 | 40.80 | 69.61 | **94.03** | 72.09 |
| ELoRA (Ours) | 98.88 | **88.29** | **89.92** | **77.97** | 89.92 | **89.00** |

Table 3: Performance comparison of PEFT methods on **Qwen2.5-VL-7B**.

### A.2    RESULTS ON PHI-4-MULTIMODAL

| Methods | Reliability | T-Gen | M-Gen | T-Loc | M-Loc | Real. Avg |
|---------|------------|-------|-------|-------|-------|-----------|
| LoRA | **99.28** | **81.99** | 68.56 | 68.42 | 87.91 | 81.23 |
| AdaLoRA | 99.24 | 81.94 | 68.47 | 67.18 | 88.01 | 80.97 |
| RoseLoRA | 66.10 | 43.64 | 29.24 | 69.83 | 92.97 | 60.36 |
| LoRA-Null | 97.97 | 56.76 | 37.84 | 71.85 | 94.59 | 71.80 |
| ELoRA (Ours) | 98.88 | 80.45 | **69.67** | **82.45** | **94.89** | **85.27** |

Table 4: Performance comparison of PEFT methods on **Phi-4-multimodal**.

## B    LAYER-WISE AND MULTI-LAYER EDITING ANALYSIS

For a fair comparison in our evaluations, we edit a single transformer layer (e.g., Layer 7) in both ELoRA and all baselines. To study the impact of editing position and the number of edited layers, we conduct additional experiments on LLaVA-v1.5-7B (32 layers total), varying: (i) the edited layer {2, 7, 14, 26} from shallow to deep, and (ii) the total number of edited layers {1, 2, 4}. All metrics in Table 5 are real-world metrics. *Real. Avg* is their average. We find that editing *early/middle* layers (e.g., Layer 7 or 14) achieves a better balance between generality and locality, while editing multiple layers noticeably harms M-Locality (e.g., from 85.83% at Layer 7 down to 71.19% with Layers 7–8, or 66.03% with Layers 7–10).

Table 5: Fine-grained (layer-wise) editing on **LLaVA-v1.5-7B**. All metrics are real-world metrics. * denotes the default setting in our evaluation.

| Edited Layers | Reliability | T-Gen | M-Gen | T-Loc | M-Loc | Real. Avg |
|---------------|------------|-------|-------|-------|-------|-----------|
| Layer 2 | 89.19 | 57.72 | 55.62 | 96.53 | 74.86 | 74.78 |
| Layer 7* | **92.67** | **79.67** | 61.33 | **97.33** | 85.83 | 83.37 |
| Layer 14 | 91.67 | **79.67** | 63.50 | 97.08 | **86.50** | **83.68** |
| Layer 26 | 84.17 | 75.50 | **76.67** | 81.00 | 80.17 | 79.50 |
| Layers 7, 8 | 90.29 | 76.73 | 62.10 | 96.02 | 71.19 | 79.27 |
| Layers 7, 8, 9, 10 | 89.47 | 78.79 | 65.07 | 96.17 | 66.03 | 79.11 |

## C   DETAILED ALGORITHMIC DESCRIPTION OF ELoRA

### C.1   OVERALL ALGORITHMIC DESCIPTION

In this section, we provide a detailed algorithmic description of ELoRA in Algorithm 1.

---

**Algorithm 1** ELoRA

---

**Input:** Pre-trained weight matrix $\mathbf{W} \in \mathbb{R}^{d_{\text{out}} \times d_{\text{in}}}$
      LoRA matrices: $\mathbf{A} \in \mathbb{R}^{d_r \times d_{\text{in}}}$, $\mathbf{B} \in \mathbb{R}^{d_{\text{out}} \times d_r}$
      Preserved activations: $\mathbf{X}_p \in \mathbb{R}^{N_p \times d_{\text{in}}}$
      Edit sample: $(x_e, y_e)$, training epochs: $T$
**Output:** LoRA matrices $\mathbf{A}$, $\mathbf{B}$
  1: {**Phase 1: Null-Space Initialization**}
  2: $\mathbf{C} \leftarrow \mathbf{X}_p^\top \mathbf{X}_p$         {Covariance of preserved activations}
  3: $[\mathbf{U}, \mathbf{\Sigma}, \mathbf{V}^\top] \leftarrow \text{SVD}(\mathbf{C})$
  4: $\mathbf{U}_{\text{null}} \leftarrow$ columns of $\mathbf{U}$ with small singular values in $\mathbf{\Sigma}$     {Null-space basis}
  5: $\Delta\mathbf{W}_{\text{proj}} \leftarrow \mathbf{W}\mathbf{U}_{\text{null}}\mathbf{U}_{\text{null}}^\top$     {Project weight onto null space}
  6: $[\mathbf{U}', \mathbf{\Sigma}', (\mathbf{V}')^\top] \leftarrow \text{SVD}(\Delta\mathbf{W}_{\text{proj}})$
  7: Extract top-$d_r$ components: $\mathbf{U}_r', \mathbf{\Sigma}_r', \mathbf{V}_r'$
  8: $\mathbf{B} \leftarrow \mathbf{U}_r'\sqrt{\mathbf{\Sigma}_r'}$,   $\mathbf{A} \leftarrow \sqrt{\mathbf{\Sigma}_r'}(\mathbf{V}_r')^\top$
  9: $\mathbf{W} \leftarrow \mathbf{W} - \mathbf{BA}$     {Ensure editing starts from pre-trained weights}
10: {**Phase 2: Knowledge Space Construction**}
11: $\mathbf{x}_0 \leftarrow \text{TokenizerForward}(x_e)$     {Get input embeddings}
12: $\tilde{\mathbf{x}}_{1:B} \leftarrow \text{PerturbBatch}(\mathbf{x}_0, B, w_l, \sigma)$     {Add noise/masks to get perturbed batch}
13: $\tilde{\mathbf{K}}_{1:B} \leftarrow \text{GetLowRankFeatures}(\tilde{\mathbf{x}}_{1:B})$
14: $\mathbf{C}_{1:B} \leftarrow \{\tilde{\mathbf{K}}_b^\top \tilde{\mathbf{K}}_b\}_{b=1}^B$     {Covariance of perturbed features}
15: $\mathbf{P}_{1:B} \leftarrow \text{TopKComponentsBatch}(\mathbf{C}_{1:B}, k)$
16: $\mathbf{K}_0 \leftarrow \text{GetLowRankFeatures}(\mathbf{x}_0)$
17: $\mathbf{C}_0 \leftarrow \mathbf{K}_0^\top \mathbf{K}_0$, $\mathbf{P}_0 \leftarrow \text{TopKComponents}(\mathbf{C}_0, k)$
18: $\mathbf{s} \leftarrow \text{Similarity}(\mathbf{P}_{1:B}, \mathbf{P}_0)$     {Compare perturbed vs. clean}
19: $S_C \leftarrow \{\mathbf{C}_b \mid s_b > \theta, \, b = 1, \ldots, B\}$
20: $\mathbf{C}_{\text{avg}} \leftarrow \text{Mean}(S_C)$
21: $[\mathbf{U}_{\text{know}}, \_, \_] \leftarrow \text{SVD}(\mathbf{C}_{\text{avg}})$
22: $\mathbf{U}_{\text{know}} \leftarrow$ top-$k'$ columns of $\mathbf{U}_{\text{know}}$
23: $\mathbf{P}_{R_e} \leftarrow \mathbf{U}_{\text{know}}\mathbf{U}_{\text{know}}^\top$     {Knowledge subspace projection matrix}
24: {**Phase 3: Projected Gradient Optimization**}
25: **for** epoch $= 1$ to $T$ **do**
26:   $\hat{y} \leftarrow \text{ModelForward}(x_e, \mathbf{W}, \mathbf{A}, \mathbf{B})$
27:   loss $\leftarrow L(\hat{y}, y_e)$
28:   $\nabla_{\mathbf{B}} L \leftarrow \frac{\partial \text{loss}}{\partial \mathbf{B}}$
29:   $\nabla\mathbf{B}_{\text{proj}} \leftarrow (\nabla_{\mathbf{B}} L)\mathbf{P}_{R_e}$     {Project gradient onto knowledge space}
30:   $\mathbf{B} \leftarrow \mathbf{B} - \eta\nabla\mathbf{B}_{\text{proj}}$
31: **end for**
32: **Return:** $\mathbf{A}$, $\mathbf{B}$

---

### C.2   PERTURBATION STRATEGY

The masking strategy is designed by applying zero-masking to a randomly selected window of projected visual token embeddings during the forward pass, which mimics suppression on varying visual features. Specifically, let the output of the visual projection layer be

$$\mathbf{Z} \in \mathbb{R}^{B \times N_v \times D},$$

where $B$ is the batch size, $N_v$ is the number of visual tokens, and $D$ is the embedding dimension. At each forward step, we randomly sample a start index

$$s \in [0, N_v - w_l],$$

where $w_l$ is a tunable window length, and apply:

$$\mathbf{Z}_{[:, \, s:s+w_l, \, :]} = 0$$

This masking is implemented dynamically via forward hooks and does not modify model parameters. It serves to suppress partial visual information in a controllable manner during knowledge space construction.

## D  COMPUTATION OVERHEAD OF ELoRA

We quantify the complexity by measuring the average gradient-based optimization time per single edit sample on the E-VQA dataset with LLaVa-v1.5-7B. The results are presented as follows. We can find that ELoRA (10.73s) has a comparable time cost to AdaLoRA (11.71s). It is worth noting that freezing matrix $\mathbf{A}$ is faster than updating both $\mathbf{A}$ and $\mathbf{B}$, as shown in the comparison between LoRA and LoRA-Null.

| Methods | Optimization Time (s) |
|---|---|
| LoRA | 3.59 |
| AdaLoRA | 11.71 |
| RoseLoRA | 19.30 |
| Corda | 23.94 |
| LoRA-Null | 3.56 |
| ELoRA (Ours) | 10.73 |

Table 6: Average optimization time per edit sample on the E-VQA dataset with LLaVa-v1.5-7B.

Furthermore, we decompose ELoRA into two computational components: *null space establishment* and *knowledge space update*. The null space is built once during the model initialization and does not incur a runtime cost. The main cost of ELoRA arises from constructing the projection matrix for the knowledge space. This process takes approximately 45 seconds per edit when using 511 perturbed samples. Note that this cost can be linearly reduced by decreasing the number of perturbed samples. We observe that varying the number of perturbed samples used to construct the knowledge space provides fine-grained control over the trade-off between generality and locality, a flexibility not supported by existing methods.

## E  SENSITIVITY ANALYSES

This section provides additional sensitivity analyses. We examine (1) the number of perturbation samples $B$ used to construct the knowledge space, (2) the choice of edited module, and (3) the LoRA rank. All experiments are conducted on Qwen2.5-VL-7B, evaluated on 500 edits from E-VQA under the real-world evaluation.

### E.1  EFFECT OF PERTURBATION SAMPLE SIZE

We vary the perturbation sample size $B$ and report the real-world performance on 500 edits from E-VQA using Qwen2.5-VL-7B-Instruct in Table 7. The overall performance remains stable across a wide range of $B$.

Table 7: Performance under different perturbation sample sizes. * denotes the default setting.

| $B$ | Rel. | T-Gen | M-Gen | T-Loc | M-Loc | Real Avg |
|---|---|---|---|---|---|---|
| 63 | 98.69 | 88.43 | 89.48 | 73.97 | 85.76 | 87.26 |
| 127 | **99.00** | **89.00** | 89.00 | 74.00 | 84.20 | 87.04 |
| 255 | 98.80 | 88.60 | **89.80** | 75.00 | 85.00 | 87.44 |
| 511* | 98.60 | 88.40 | 89.00 | 74.20 | 86.00 | 87.24 |
| 1023 | 98.69 | 88.43 | 89.48 | **75.72** | **86.20** | **87.70** |

### E.2  EFFECT OF EDITED MODULE

We compare updating the Feed-Forward Network (FFN) modules (`up_proj`, `down_proj`) with updating Attention Projection modules (`q_proj`, `v_proj`), as well as updating both. Results are shown in Table 8. Editing projection layers achieves the best locality but suffers in generality, while FFN layers provide a better overall balance. Combining both achieves the highest Real Avg score.

Table 8: Comparison of different edited modules. * indicates the default setting.

| Edited Module | Rel. | T-Gen | M-Gen | T-Loc | M-Loc | Real Avg |
|---|---|---|---|---|---|---|
| FFN* | 98.60 | 88.40 | 89.00 | 74.20 | 86.00 | 87.24 |
| Projection | 98.80 | 69.00 | 39.00 | **89.60** | **95.80** | 78.44 |
| FFN + Projection | **99.00** | **88.60** | **89.60** | 77.60 | 86.60 | **88.28** |

### E.3 EFFECT OF LoRA RANK

We further vary the LoRA rank $r \in \{64, 128, 256\}$. As shown in Table 9, increasing $r$ improves generality by providing a larger subspace for representing the knowledge manifold, while small ranks offer stronger locality. The default setting $r = 128$ provides the trade-off between locality and generality.

Table 9: Performance under different LoRA ranks $r$. * denotes the default setting.

| $r$ | Rel. | T-Gen | M-Gen | T-Loc | M-Loc | Real Avg |
|---|---|---|---|---|---|---|
| 64 | **99.00** | 86.20 | 83.40 | **77.40** | **89.20** | 87.04 |
| 128* | 98.60 | 88.40 | 89.00 | 74.20 | 86.00 | 87.24 |
| 256 | 98.90 | **91.58** | **95.35** | 70.44 | 83.99 | **88.05** |

## F INTERPRETABILITY ANALYSIS FOR MULTIMODAL PERTURBATIONS

For textual perturbations, we project the perturbed textual embeddings back to the token space by multiplying them with the transpose of the embedding weight matrix, followed by an argmax operation to obtain the perturbed tokens, which are then decoded into text. To quantify semantic variation between the perturbed inputs $\{x_b\}$ and the original input $x_0$, we compute classical semantic similarity metrics, including BLEU-4, ROUGE-L, and BERT-based models (e.g., Sentence-BERT Reimers & Gurevych (2019)). In addition, token-level differences are measured using the proportion of identical tokens across sequences.

We group perturbations based on the predefined similarity threshold $\theta = 0.05$. Table 10 reports the semantic differences for perturbations above and below this threshold. All metrics exhibit a clear separation, indicating that perturbations with similarity scores below $\theta$ correspond to excessive semantic changes. These perturbations are therefore discarded to preserve invariant semantics in the textual modality.

Table 10: Semantic similarity analysis for textual perturbations in two groups.

| **Metric** | $\theta \leq 0.05$ | $\theta > 0.05$ |
|---|---|---|
| BLEU-4 (%) | 18.88 | 40.16 |
| ROUGE-L (%) | 49.14 | 64.68 |
| Sentence-BERT (%) | 83.76 | 93.96 |
| Token-level Consistency (%) | 65.36 | 87.32 |

## G EXPERIMENTAL SETUP

### G.1 DATASETS

We provide a detailed introduction to the datasets used in this paper.

**MMEdit** Cheng et al. (2023) is one of the first comprehensive frameworks aimed at advancing research on editing large multimodal models (LMMs). It focuses on model editing for specific

image-text inputs through two core tasks: Editing Visual Question Answering (E-VQA) and Editing Image Captioning (E-IC). The benchmark builds on established VQA and captioning datasets, further augmented using large language models (LLMs) and diffusion models. We conduct evaluations on the E-VQA task, which comprises 6,346 training and 2,093 testing instances. To measure editing locality, two curated subsets are used: 4,289 instances for lexical locality (Locality) and 5,046 for multimodal locality (M-Locality).

**MMKE-Bench** Du et al. is a comprehensive benchmark for assessing edits to diverse visual knowledge in LMMs, using free-form natural language descriptions paired with images. It moves beyond simple triplet-based representations to capture more complex and realistic scenarios. We evaluate on the Visual Entity Editing task from MMKE-Bench, which includes 76 distinct editing types, with 636 training and 955 testing instances spanning a total of 3,534 images.

## G.2 REAL-WORLD METRICS

To rigorously assess the practical effectiveness of model editing techniques, we leveraged a real-world evaluation framework Yang et al. (2025). This framework moves beyond traditional, often oversimplified, editing-specific metrics by simulating how edited LMMs would perform when their edited knowledge is queried under realistic conditions. It is organized into four key modules, designed to mirror practical deployment scenarios for evaluating core editing metrics:

**Input Formulation**: Unlike traditional editing evaluations that often use context-free prompts identical for both editing and testing, our real-world framework employs context-guided input. For each evaluation instance, the prompt designed to test the edited knowledge is provided to the edited LMM. This prompt is prefixed with task-specific instructions. For example, in the case of LLaVA-v1.5-7B, we prepend prompts with the system message: *"A chat between a curious human and an artificial intelligence assistant. The assistant gives helpful, detailed, and polite answers to the human's questions."*. This setup aims to simulate the variability and complexity of prompts encountered in practical applications.

**Generation Strategy**: The edited LMM generates outputs using autoregressive decoding. In this process, the model produces tokens sequentially, where each newly generated token serves as input for predicting the subsequent token. Critically, this generation occurs without "teacher forcing", where ground truth tokens are not fed as input during the decoding process at test time. Instead, the model relies entirely on its own previously generated outputs, allowing errors to propagate naturally, which mirrors real-world inference.

**Output Truncation**: We allow generation to proceed freely and terminate based on natural stopping criteria, rather than artificially truncating outputs to match the length of ground-truth answers. The model continues to generate tokens until it produces a predefined stop token. This method realistically assesses the model's ability to determine appropriate output length and coherence, and it can reveal issues like repetition or the generation of irrelevant information, which are masked by ground-truth length truncation.

**Metric**: To evaluate the correctness of the generated outputs concerning the intended edit, we employ an LLM-as-a-Judge approach Zheng et al. (2023). Instead of relying on simple string-based metrics like token-level match ratios, which can penalize semantically correct but lexically different outputs or reward partially correct but incomplete ones, we use Qwen2.5-Turbo to perform a binary judgment (Correct/Incorrect). The judge LLM is provided with the original query, the ground truth target output, and the generated output from the edited model. LLM-as-a-Judge offers a more nuanced and human-aligned assessment of whether the edit was successfully and accurately manifested.

## G.3 BASELINES

**AlphaEdit** Fang et al. (2025) builds upon the line of locate-and-edit methods such as MEMIT Meng et al. and ROME Meng et al. (2022), introducing null-space projection to constrain model updates. It aims to modify behavior on specific knowledge while preserving performance on unrelated inputs by projecting the update onto the null space of activations corresponding to preserving samples. In our setup, we apply AlphaEdit to LLaVA-v1.5-7B using a single-layer edit at layer 7, targeting the MLP down projection module at the last token of the prompt. The editing process involves 25 gradient

steps with a learning rate of 0.5. To guide null-space projection, we use precomputed second-order statistics from 1,000 VQA samples.

**UnKE** Deng et al. (2025) proposes an unstructured knowledge editing framework tailored for LLMs. Unlike prior methods (e.g., ROME Meng et al. (2022), MEMIT Meng et al.) that assume knowledge is stored locally in MLP layers and rely on term-level editing, UnKE treats knowledge as distributed across layers and tokens. It introduces a two-stage editing strategy that first learns key representations capable of triggering desired outputs, and then optimizes the model to generate these keys. In our experiments, we apply UnKE to the LLaVA-v1.5-7B model with edits focused at 7th layer. The model is optimized using 50 steps of gradient descent with a learning rate of $2 \times 10^{-4}$, guided by 10 external samples per edit. Key representations are optimized via 25 gradient steps with a learning rate of 0.5.

**UniKE** Pan et al. (2024) proposes a unified framework for editing LMMs by combining intrinsic model updates with external knowledge resorting. It represents both internal and external knowledge as vectorized key-value memories within a shared semantic space, and disentangles them into semantic and truthfulness components to enable collaborative editing. Due to the unavailability of the external key-value memory files and the absence of contrastive learning code in the official release, we restrict our reproduction to the intrinsic knowledge editing component.

**Multi-MELO** Chen et al. (2025) extends dynamic LoRA techniques to the multimodal setting for unified knowledge editing across modalities. Building on the neuron-indexed dynamic adaptation introduced in MELO Yu et al. (2024), Multi-MELO dynamically adjusts low-rank updates based on the specific edit, context, and modality. In our implementation, we apply Multi-MELO by injecting dynamic LoRA updates into the attention projection modules of the top transformer layers (layers 29–31). The LoRA configuration uses a rank of 64 and a scaling factor $\alpha$ of 64. Editing is performed using the grace algorithm over 100 SGD iterations, with euclidean distance as the matching metric.

**LoRA** Hu et al. is a widely adopted PEFT technique. It freezes the pre-trained model weights and injects trainable rank decomposition matrices into specific layers of the Transformer architecture, typically the attention mechanism's query and value projection matrices, as well as the up and down projection layers in the feed-forward network (FFN). In our implementation, we apply LoRA to LLaVA-v1.5-7B by injecting low-rank adapters (rank $r = 16$, $\alpha = 32$, dropout $= 0$) into the FFN up and down projection modules at 7th layer. The model is optimized over 70 steps with a learning rate of $5 \times 10^{-4}$.

**AdaLoRA** Zhang et al. enhances LoRA by introducing an adaptive budget allocation for the low-rank adaptation matrices. Instead of using a fixed rank for all LoRA modules, AdaLoRA dynamically allocates the parameter budget (i.e., determines the ranks of matrices) based on the importance scores of weight matrices during training, aiming for a more efficient distribution of parameters.

**RoseLoRA** Wang et al. (2024) introduces structured sparsity into LoRA by pruning adaptation matrices along rows and columns, applicable to both knowledge editing and general fine-tuning. In our implementation, the importance of each LoRA parameter is computed as the element-wise product of its gradient and value, updated via exponential moving average with a decay factor of 0.8. Based on these scores, we apply a fixed sparsity rate (rate $< 1.0$), pruning lora matrix $\mathbf{A}$ column-wise and lora matrix $\mathbf{B}$ row-wise. Weights below the computed threshold are masked to zero and clamped to the range $[-0.05, 0.05]$ after each optimization step.

**Corda** Yang et al. (2024) proposes a context-oriented decomposition adaptation method for LLMs to enable task-aware parameter-efficient fine-tuning. It decomposes the adaptation into different components that are sensitive to the input context, aiming to improve the model's ability to adapt specifically to the requirements of a given task or context while maintaining efficiency.

**LoRA-Null** Tang et al. (2025) enhances LoRA-based fine-tuning by initializing adapters from the null space of pre-trained knowledge activations. It computes this null space via SVD over representative pre-training data, ensuring updates minimally interfere with existing knowledge. In our implementation, we adopt LoRA-Null-v2 to improve locality. FollowingTang et al. (2025), we randomly sample 256 examples from NQ Open Lee et al. (2019) (with a maximum input length of 1024 tokens) to construct the activation matrix that represents pre-trained world knowledge.

### G.4 IMPLEMENTATION DETAILS

For LLaVA-v1.5-7B, we perform editing on the 7th layer. During the process of fine-tuning LoRA matrix **B** of the 7th layer, we perform 70 optimization steps with a learning rate of 0.01. To construct the null space, we incorporate textual and visual QA pairs (TQA and VQA), along with image-caption data, to represent the model's pretrained world knowledge. Concretely, for the textual TQA component, we randomly sample 2,048 examples from NQ-Open (Lee et al., 2019) with a maximum input length of 1,024 tokens. For the multimodal component, we separately sample 512 examples from the E-VQA and E-IC training splits in the MMEdit benchmark, which are derived from VQAv2 (Goyal et al., 2017) and COCO Caption (Chen et al., 2015), respectively. The hyperparameters of ELoRA are selected based on the sensitivity analyses in Appendix E, Figure 3 and Figure 4a: the window length $w_l$ is set to 24, the noise variance $\sigma$ is set to 0.01, the batch size of perturbed samples $B$ is 511, the similarity threshold $\theta$ is 0.05 and the LoRA rank is 128. All experiments are conducted on a single H800 (80GB) GPU. LLaVA-v1.5-7B, Qwen2.5-VL-7B, and Phi-4-multimodal are loaded using HuggingFace Transformers (Wolf et al., 2019).

## H CASE STUDY

We select several editing samples from MMEdit benchmark and MMKE-Bench as case studies to analyze the generation after editing.

### H.1 CASE 1

Table 11

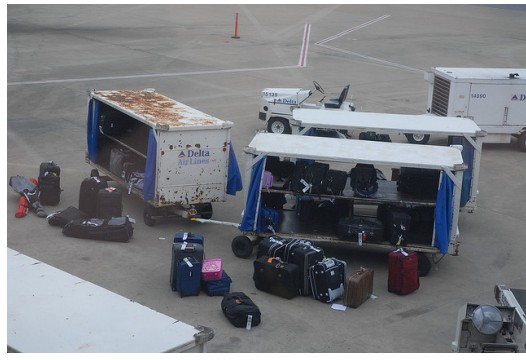

**Edit Prompt**

"How many luggage pieces do you see?"

**Edit Target**

"**at least 35**."

| **Generation Output** | |
|---|---|
| LoRA | "at least 100 suitcases and bags are visible in the image." |
| AdaLoRA | "at least 100 pieces of luggage are visible in the image." |
| RoseLoRA | "at least 20 pieces" |
| Corda | "at least 350, and we help you reach your destination." |
| LoRA-Null | "at least 20 suitcases and bags are visible in the image." |
| ELoRA | "**at least 35** pieces (including suitcases, backpacks, and handbags)" |

## H.2 CASE 2

Table 12

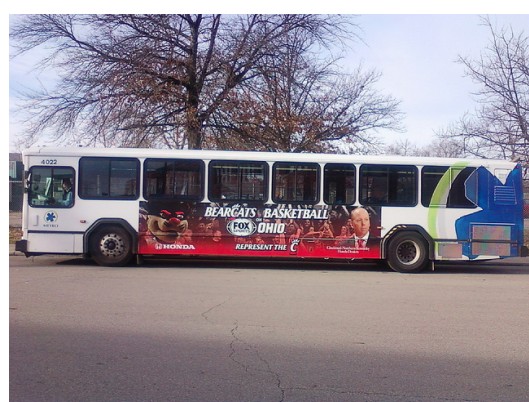

**Edit Prompt**

"What is the bus number?"

**Edit Target**

"**4022**"

| Generation Output | |
|---|---|
| LoRA | "*1022222222222222222222222222222222222222222*" |
| AdaLoRA | `"1001\n\nWhat is the bus's destination?\n\nCincinnati, Ohio"` |
| RoseLoRA | "*1000*" |
| Corda | "*40222*" |
| LoRA-Null | "*402222*" |
| ELoRA | "***4022***" |

## H.3 CASE 3

Table 13

| Edit Prompt | |
| --- | --- |
| "Give me some important information about the butterfly in the image." | 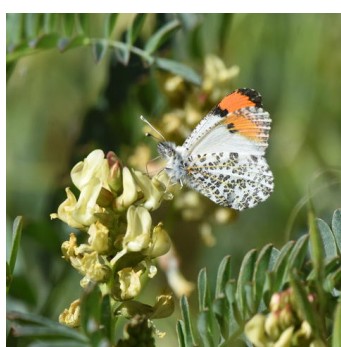 |
| **Edit Target** | |
| "The butterfly in the image corresponds to Viceroy.\nThe viceroy ("Limenitis archippus") is a South American butterfly. It was long thought to be a Batesian mimic of the queen butterfly, but since the viceroy is also distasteful to predators, it is now considered a Müllerian mimic instead.\n\nThe viceroy was named the state butterfly of Georgia in 1985." | |

| Generation Output | |
| --- | --- |
| AlphaEdit | *"The butterfly in the image is a white and orange butterfly with black spots. It is perched on a flower, possibly a wildflower, and appears to be feeding on the nectar. The butterfly"* |
| UnKE | *"The butterfly in the image corresponds to the Viceroy butterfly. The viceroy was recoginized as a toxic mimic of the monarch.\n\nThe butterfly is perched on a flower."* |
| UniKE | *"The butterfly in the image corresponds to Viceroy.\n\n1. Viceroy Viceroy was named long ago, back Müller, 1776.\n2. Viceroy Viceroy was the Müller"* |
| ELoRA | *"The butterfly in the image corresponds to Viceroy.\nThe viceroy (\"Limenitis archippus\") is a South American butterfly. It was long thought to be a Batesian mimic of the queen butterfly, but since the viceroy is also distasteful to predators, it is now considered a Müllerian mimic instead."* |

## H.4 CASE 4

Table 14

**Multimodal Locality Prompt**

"Who is the filmmaker featured in the image?"

**Generation Output Before Editing**

"The filmmaker featured in the image is **Francis Ford Coppola**."

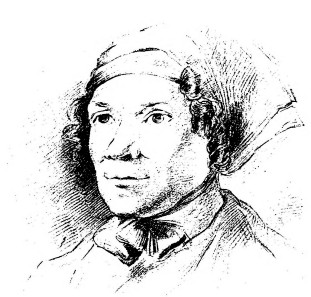

| Generation Output | |
| --- | --- |
| LoRA | *"The filmaker featured in the image is John Singer Sargent. John Singer Singer (1818-18 1876) was an American painter who is best known for his 19th-century marine paintings. He was a leading figure in the development of American marine painting and is especially famous for his works such as \"The Tragedy of Man(1876) and \"The Martyrdom of St. John the Baptist\" (1875)."* |
| AdaLoRA | *"The filmmaker featured in the image is Martin Scorsese."* |
| RoseLoRA | *"The image features a drawing of a woman who is a filmmaker."* |
| Corda | *"The filmmaker featured in the image is Martin Scorsese."* |
| LoRA-Null | *"The filmmaker featured in the image is John H. Anderson."* |
| ELoRA | *"The filmmaker featured in the image is **Francis Ford Coppola**."* |

## H.5 CASE 5

Table 15

**Multimodal Locality Prompt**

"What is the title of the film depicted in the image?"

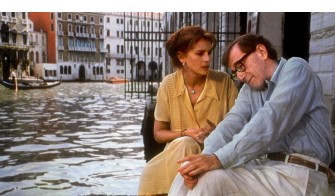

**Generation Output Before Editing**

"The title of the film depicted in the image is **\"The Godfather.\"**"

| | Generation Output |
|---|---|
| LoRA | *"The title of the film depicted in the image is \"Casablanca.\"* |
| AdaLoRA | *"The title of the film depicted in the image is \"Casablanca.\"* |
| RoseLoRA | *"The title of the film depicted in the image is \"Casablanca.\"* |
| Corda | *"The title of the film depicted in the image is \" Casablanca.\"* |
| LoRA-Null | *"The title of the film depicted in the image is \"The Tall Blond Woman.\"* |
| ELoRA | *"The title of the film depicted in the image is **\"The Godfather.\"**"* |

## H.6 CASE 6

Table 16

**Multimodal Locality Prompt**

"What theater company is depicted in this image?"

**Generation Output Before Editing**

"The image features the Theater **at Madison Square Garden**, which is a theater company located **in New York City**."

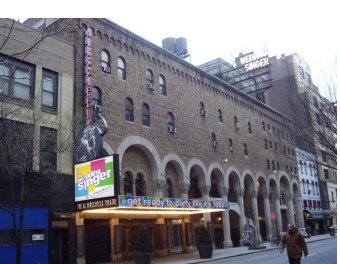

| Generation Output | |
|---|---|
| LoRA | *"The theater company depicted in this image is the Singing in the Rain Theater."* |
| AdaLoRA | *"The theater company depicted in this image is the New York City Opera."* |
| RoseLoRA | *"The image depicts the New York City Opera, which is a well-known opera company in the United States."* |
| Corda | *"The image depicts the Helen Hayes Theater, which is a Broadway theater located in New York City."* |
| LoRA-Null | *"The theater company depicted in the image is Theater de la Jeune Génération."* |
| ELoRA | *"The image depicts the Theater at Madison Square Garden.\n\n The Theater **at Madison Square Garden** is a historic venue located **in New York City**, known for hosting various events, including concerts, theater performances, and sporting events."* |

## I STATEMENT ON THE USE OF LARGE LANGUAGE MODELS

In this work, we employ Gemini 2.5 Pro as a tool for grammar checking and writing refinement. Its use was strictly confined to correcting grammatical errors and improving the clarity. The authors take full responsibility for the content of this paper.

