# OpenReview forum: "Disentangling Multimodal Knowledge Preservation and Editing via Low-rank Adaptation"
_ICLR.cc/2026/Conference — Submitted to ICLR 2026_

### Official Review · Reviewer_92Un · 2025-10-25

**Soundness:** 2
**Presentation:** 3
**Contribution:** 2
**Rating:** 4
**Confidence:** 4

**Summary:**

The paper proposes ELoRA, a LoRA-based framework that decomposes the parameter update space into a null space for preserving existing knowledge and a knowledge space for injecting new information, aiming to balance locality and generality in multimodal editing.

**Strengths:**

The paper is clearly written and well-organized, with fluent language that make the presentation easy to follow.

**Weaknesses:**

1.	The authors claim that directly leveraging LoRA for multimodal KE faces challenges, including low locality and low generality. However, these two challenges do not appear to be specific to multimodal KE, and the paper does not clearly differentiate them from those already existing in unimodal KE.
2.	The title emphasizes mitigating the challenges of multimodal knowledge editing, yet the proposed method does not seem to provide further insights or methodological innovations from a multimodal perspective.
3.	The proposed null space idea is highly related to AlphaEdit [1], and the initialization strategy also resembles that of CorDA [2], which substantially reduces the method’s novelty. The authors should clarify the differences and additional contributions compared to these two methods.
4.	The authors introduce Gaussian noise and other perturbations to construct the knowledge space, but the robustness of this approach is questionable. Could such perturbations risk losing important information or even induce additional hallucinations?
5.	In Fig. 1 (a), the colors of the different lines are too similar, making them difficult to distinguish.

[1] AlphaEdit: Null-Space Constrained Knowledge Editing for Language Models

[2] CorDA: Context-Oriented Decomposition Adaptation of Large Language Models for Task-Aware Parameter-Efficient Fine-tuning

**Questions:**

1.	As noted in the weaknesses, could the authors elaborate on the differences and unique contributions of their method compared with AlphaEdit, LoRA-Null, and CorDA?
2.	Why does AlphaEdit perform significantly worse on the MMKE-Bench dataset? Could the authors provide an analysis or explanation for this phenomenon?
3.	Since the paper emphasizes multimodal knowledge editing, could the authors clarify which parts of the proposed method are explicitly designed to handle multimodal data or interactions?

---

> ### Author Response · Authors · 2025-11-21
> **Response to Reviewer 92Un (Part 1)**
>
> Thank you for your effort and time devoted to reviewing our submission. We appreciate the opportunity to clarify our method's novelty and its specific contributions to the multimodal knowledge editing community.
>
> ## Challenges (locality/generality) are not specific to multimodal knowledge editing.
>
> >The authors claim that directly leveraging LoRA for multimodal KE faces challenges, including low locality and low generality. However, these two challenges do not appear to be specific to multimodal KE, and the paper does not clearly differentiate them from those already existing in unimodal KE.
>
> We agree that locality and generality are also considered in unimodal knowledge editing (KE). However, experiments in Table 1 indicate that the challenge of achieving reliability, generality, and locality simultaneously becomes more severe in multimodal KE, **creating a "trilemma" that unimodal methods fail to solve**. We further give three explanations:
>
> - **Reliability becomes harder due to longer and more variable multimodal inputs.** Multimodal inputs combine image tokens and text tokens, which substantially increases sequence length and the complexity of editing objectives. As shown in Table 1, even strong unimodal editors (ROME, MEMIT, AlphaEdit) experience degraded reliability when applied to E-VQA, whereas base LoRA maintains higher reliability.
> - **Locality becomes more fragile due to the cross-modal preservation requirement.** LMMs must preserve knowledge in both text-only and image-conditioned inputs that are irrelevant to new edits. As shown in Table 1, although base LoRA retains relatively strong reliability compared with unimodal KEs, it suffers a much sharper drop in M-Loc (multimodal locality) compared to T-Loc (textual locality), proving that visually grounded knowledge is more fragile and entangled than text-only knowledge. Consequently, directly applying LoRA to multimodal KE is not viable.
> - **Generality is intricately shaped by dual perturbation sources: visual variations and textual paraphrases.** In unimodal KE, generality implies robustness to textual paraphrases. In multimodal KE, it requires robustness to visual variations (e.g., different angles, lighting, or occlusions of the same object).
>
>
> **Furthermore, these difficulty becomes even clearer on MMKE-Bench**, which contains open-ended, unstructured multimodal knowledge and diverse visual conditions, making it substantially more difficult than triplet-based and fact-structured datasets. As evidenced in Table 1, all unimodal methods, including the unstructured knowledge editor UnKE, perform poorly on MMKE-Bench, confirming that approaches designed for unimodal knowledge cannot generalize to real-world multimodal edits.
>
> **In contrast, the proposed ELoRA addresses the above three challenges by** (1) adopting a LoRA-based mechanism to improve editing reliability, (2) preserving knowledge in a dedicated multimodal null space, and (3) designing a complementary multimodal knowledge space by introducing modal-specific variance to enhance generality.
>
> ## Why does AlphaEdit perform poorly on MMKE-Bench?
>
> >Why does AlphaEdit perform significantly worse on the MMKE-Bench dataset? Could the authors provide an analysis or explanation for this phenomenon?
>
> Thank you for the question. The success of ROME and MEMIT on editing atomic facts through rank-one (i.e., one-dimensional) updates suggests that simple factual knowledge is often localized in extremely low-dimensional linear subspaces. UnKE extends knowledge editing from structured triplets to unstructured textual knowledge, and its previous experiments [1] already show that ROME and MEMIT perform poorly in such settings. **While AlphaEdit is a remarkable extension of MEMIT with the addition of a null-space constraint, it still inherits the same limitation: rank-one updates are insufficient for modifying complex, unstructured knowledge.**
>
> **For multimodal knowledge editing, the challenge becomes even more severe.** The E-VQA task resembles editing structured multimodal facts, although the dataset lacks explicit subject tokens. As reported in Table 1, ROME, MEMIT, and AlphaEdit all achieve lower reliability than even base LoRA. So we hypothesize that multimodal knowledge is encoded not in a one-dimensional direction, but in a low-dimensional manifold (please see Section 3.4).
>
> **MMKE-Bench consists of unstructured and multimodal knowledge, making the inadequacy of a rank-one update even more evident.** Combining the previous observations from UnKE [1] and our experiments, it is therefore expected that AlphaEdit performs worse on MMKE-Bench.
>
> To the best of our knowledge, we are the first to reimplement ROME, MEMIT, AlphaEdit, and UnKE in the multimodal knowledge editing setting. We will release our implementations to facilitate further analysis and benchmarking within the knowledge editing community.

---

> ### Author Response · Authors · 2025-11-21
> **Response to Reviewer 92Un (Part 2)**
>
> ## Methodological innovations from a multimodal perspective.
>
> >Since the paper emphasizes multimodal knowledge editing, could the authors clarify which parts of the proposed method are explicitly designed to handle multimodal data or interactions?
>
> Thank you for the question. We emphasize that the core methodological innovation of ELoRA does not lie in the null space alone, but in the multimodal decomposition of the editing space into a null space and a complementary knowledge space, which is fundamentally different from prior work and necessary for multimodal knowledge editing.
>
> First, prior null-space LoRA methods such as LoRA-Null and AlphaEdit only preserve knowledge, but **they inherently restrict the update direction and therefore fail to support editing generality**. As shown in Table 1, both methods maintain locality but exhibit limited generality, even falling below base LoRA in multimodal scenarios. This demonstrates that simply applying a null-space constraint is insufficient for multimodal KE.
>
> **In contrast, our contribution lies in introducing a complementary knowledge space constructed from modality-specific perturbed embeddings.** The combination of null space and knowledge space is **not a mechanical aggregation but a deliberate design** that simultaneously preserves multimodal knowledge and enables a controlled, generalizable update direction. As illustrated in **Figure 4(b)**, the resulting activations form two modality-specific sub-clusters, and the principal editing direction extracted from these samples spans both clusters. This property is not observed in unimodal KE, and it provides the foundation for cross-modal generalizable edits.
>
> In conclusion, ELoRA is a necessary design to resolve the conflict between preserving existing knowledge and enabling generalizable edits in LMMs. This is evidenced by the significant average performance gains it has achieved compared to the aforementioned null-space methods: LoRA-Null (e.g., +11.47% on E-VQA and +24.84% on MMKE-Bench under real-world evaluation) and AlphaEdit (e.g., +22.04% on E-VQA and +49.95% on MMKE-Bench under real-world evaluation).
>
>
>
>
> ## On the differences from AlphaEdit, LoRA-Null, and CorDA.
>
> > As noted in the weaknesses, could the authors elaborate on the differences and unique contributions of their method compared with AlphaEdit, LoRA-Null, and CorDA?
>
> We appreciate the opportunity to clarify our position. Our contributions are distinct in terms of paradigm, architecture, and task applicability:
>
> - **AlphaEdit** introduces the null space technique into knowledge editing and follows **locate–then–edit** paradigm as MEMIT, but does not incorporate multimodal structure. The null-space constraint is a standard tool explored in several prior works, **our contribution lies in how we construct and combine null and knowledge spaces for multimodal editing**. In our experiments, AlphaEdit shows strong locality but limited reliability and generality.
>
> - **LoRA-Null** focuses on avoiding catastrophic forgetting in downstream task fine-tuning, **a setting where there is no need to construct a knowledge space**. In contrast, ELoRA is specifically designed for the knowledge editing scenario, where only one or a few examples are available, making it essential to identify an invariant semantic manifold to support generalizable edits. Consistent with this distinction, all experiments show that while LoRA-Null preserves locality, it exhibits limited generality.
>
> - **CorDA** performs context-aware LoRA initialization and is designed to improve downstream task performance rather than preserve pretrained knowledge or enhance editing capability. It does not define a null/knowledge space, and in our evaluations, it shows limited performance in knowledge editing.
>
> **Summary of our contribution:** ELoRA is the first framework to disentangle the editing objective into two orthogonal subspaces: a multimodal null space (for locality) and a multimodal knowledge space (for generality). This specific decomposition is necessary to solve the "trilemma" in multimodal editing, a problem that none of the aforementioned methods can solve individually.
>
> ## On the color similarity in Fig. 1(a).
>
> >In Fig. 1 (a), the colors of the different lines are too similar, making them difficult to distinguish.
>
> We appreciate this suggestion and have revised Figure 1(a) with a higher-contrast color to improve visual clarity.

---

> ### Author Response · Authors · 2025-11-21
> **Response to Reviewer 92Un (Part 3)**
>
> ## On the robustness of Gaussian and masking perturbations for constructing the knowledge space.
>
> >The authors introduce Gaussian noise and other perturbations to construct the knowledge space, but the robustness of this approach is questionable. Could such perturbations risk losing important information or even induce additional hallucinations?
>
> The review's concern focuses on whether perturbation–based editing may introduce semantic drift or hallucinations. **ELoRA addresses the risks through two designs:** (1) perturbations are used only to probe variations in the model’s internal states rather than as training signals, preventing overfitting to superficial noise, and (2) a principled filtering mechanism selects only those perturbations that preserve semantic invariance. Details are as follows.
>
> **(1) First, ELoRA does not optimize on perturbed samples themselves.** Instead, perturbations are only used to reveal variations in the model’s internal states (please see Section 3.4), from which we extract a direction that reflects the shared underlying semantic concept. This is different from fitting to noisy inputs. To validate this design, **we conducted an additional comparison with a baseline that updates the model by directly fine-tuning on the perturbed samples**. As shown below (500 edits on E-VQA with Qwen2.5-VL-7B), directly training on perturbed samples yields higher T-Loc but severely harms both T-Gen and M-Gen, indicating poor generality.
>
> | Method   | Reliability | T-Gen | M-Gen | T-Loc | M-Loc | Real. Avg |
> |----------|-------------|-------|-------|-------|-------|-----------|
> | Baseline | 98.25       | 67.75 | 33.00 | 83.50 | 88.75 | 74.25 |
> | ELoRA    | 98.75   | 87.75 | 77.50 | 77.75 | 90.25 | 86.00 |
>
>
>
> **(2) Second, the filtering mechanism (see Eq.9) discards perturbations that deviate far from the original semantics. We conducted an additional interpretability analysis (newly added to Section 4.3) to verify this:**
>
> - **For the textual modality**, we map perturbed textual embeddings back to tokens by multiplying them with the transpose of the embedding weight matrix and taking the argmax, and then decode the tokens into texts. We then measure semantic deviation between perturbed inputs $\{x_b\}$ and $x_0$ using BLEU-4, ROUGE-L, and BERT-based models to extract sentence embeddings, along with token-level consistency. Perturbations are grouped by the similarity score threshold $\theta=0.05$. Across all metrics, perturbations above the threshold show higher semantic consistency, which validates that the filtering mechanism effectively excludes variants that diverge from the intended meaning.
>
>     |               | $\theta \leq 0.05$ | $\theta > 0.05$ |
>     | ------------- | :------------------: | :---------------: |
>     | BLEU-4 (%)       |        18.88            |    40.16            |
>     | ROUGE-L (%)      |      49.14              |    64.68          |
>     | Sentence-BERT (%)  |      83.76            |     93.96          |
>     | Token-level Consistency (%)         |  65.36                  |      87.32           |
>
>
> - **For the visual modality**, **we further evaluate whether perturbations preserve key visual subjects.** We visualize the impact of perturbed visual embeddings on the image as shown **in the newly added Figure 5** in the revised manuscript. We visualize the effect of perturbed visual embeddings using a gradient-weighted reconstruction method similar to Grad-CAM. Concretely, we attribute the L2 distance between $\mathbf{Z}_b$ (denoted in Appendix C.2) and the original embedding $\mathbf{Z}_0$ to the $24\times24$ patch grid, and then upsample the attribution map to the full $336\times336$ resolution. As shown in the newly added Figure 5, perturbations with small similarity scores often correspond to clear disruptions of key objects (e.g., the tennis racket or tennis ball), undermining the semantics relevant to the edit (e.g., "How many tennis balls are in the picture?"). This further supports that the filter removes semantically disruptive samples.
>
> **(3) Finally, we examine the robustness of the perturbation hyperparameters.** All key hyperparameters were selected based on sensitivity studies shown in Figure 3 and Figure 4a. In particular, the similarity threshold $\theta$ controls which perturbed samples contribute to the knowledge space. As shown in Figure 4a (right), including overly noisy perturbations (e.g., $\theta=0$) degrades both locality and generality, while moderate filtering ($\theta=0.05$) achieves the best balance. For the perturbation parameters, Figure 3 shows that $w_l=24$ and $\sigma=0.01$ provide sufficient diversity without harming semantic invariance, and Figure 4a (left) further indicates that varying $w_l$ has minimal effect on final performance. These settings are consistently applied across LLaVA-v1.5-7B, Qwen2.5-VL-7B, and Phi-4-multimodal.

---

> ### Author Response · Authors · 2025-11-27
> **A Gentle Reminder as We Approach the Final Week.**
>
> Dear Reviewer 92Un,
>
> Thank you again for your thoughtful review and feedback.
>
> As the rebuttal period is coming to an end, we would like to kindly confirm whether our previous response has addressed your primary concerns. We also hope the additional interpretability analyses we provided were helpful in clarifying our contributions. If there are still points that you feel require further explanation, we would be glad to provide a deeper explanation and further supporting details.
>
> We sincerely appreciate your constructive feedback and consideration.
>
> Best regards,
>
> Submission3972 Authors

---

### Official Review · Reviewer_og9B · 2025-10-28

**Soundness:** 3
**Presentation:** 2
**Contribution:** 3
**Rating:** 6
**Confidence:** 4

**Summary:**

This paper proposes ELoRA, which achieves precise and targeted updates for multimodal models. The method projects the A matrix in LoRA to a null space aligned with preserved knowledge, and projects the B matrix to a knowledge space extracted from perturbed internal states, in order to maintain pre-trained knowledge and generalize new knowledge during editing.

**Strengths:**

- The paper proposes a new editing method ELoRA for multimodal models, addressing the challenge of simultaneously preserving pre-trained knowledge and generalizing new knowledge during editing.
- The authors considered the problem that using a single edit sample can easily cause overfitting, and solved this issue through data augmentation by adding Gaussian noise to the samples.
- The authors effectively combined the advantages of editing methods and LoRA methods.

**Weaknesses:**

- ELoRA successfully transferred the application of null space from editing methods to LoRA, but I think this still seems like a combination of two methods, with somewhat limited innovation.
- The paper lacks introduction to some experimental settings, such as what r is used in ELoRA and which module in the 7th layer is the object being edited.
- The paper lacks discussion on the editing targets. As a LoRA-like method, ELoRA should be applicable to every module in the model (FFN or Attention), but the paper does not discuss whether there would be differences when various modules serve as editing targets.

**Questions:**

- What r is used in ELoRA?
- Which module is the object being edited?

---

> ### Author Response · Authors · 2025-11-21
> **Response to Reviewer og9B**
>
> Thank you for your positive assessment and your recognition of our well-motivated methods. We address your concerns as follows.
>
> ## On how ELoRA differs from directly applying null-space constraints within LoRA.
>
> >ELoRA successfully transferred the application of null space from editing methods to LoRA, but I think this still seems like a combination of two methods, with somewhat limited innovation.
>
> We wish to clarify that ELoRA represents a novel decomposition framework designed to resolve the specific conflict between locality and generality, rather than a simple combination of existing methods.
>
> Table 2 provides direct empirical evidence that a simple combination is insufficient:
>
> 1.  **Null space alone is restrictive ($M_2$ vs. $M_0$):** As shown in Table 2, simply applying the null space constraint ($M_2$) significantly improves multimodal locality (M-Loc: $35.77\% \to 84.62\%$). However, this comes at a cost: multimodal generality (M-Gen) actually drops compared to Base LoRA ($54.52\% \to 47.54\%$). This confirms that **the null space overly constrains the update**, preventing the model from generalizing to visual variations.
> 2.  **Knowledge space unlocks generality ($M_5$ vs. $M_2$):** It is the introduction of our novel knowledge space ($M_5$) that resolves this bottleneck. By projecting the update matrix $B$ onto this semantic manifold, M-Gen is recovered and significantly boosted ($47.54\% \to 63.12\%$), while maintaining high locality.
>
> To address this, **we are the first to disentangle knowledge preservation and updating in LMMs by decomposing the editing space into two orthogonal components**. In conclusion, ELoRA is not merely a technical extension, **but a necessary design** to enable null-space constraints to coexist with high editing generality. This decomposition yields substantial performance gains over prior null-space methods, such as LoRA-Null (+24.84% on MMKE-Bench) and AlphaEdit (+49.95% on MMKE-Bench).
>
> ## Regarding experimental settings and discussion on editing targets.
>
> In our main experiments, we set the LoRA rank to $r=128$ and targeted the Feed-Forward Network (FFN) modules (specifically `up_proj` and `down_proj`) in the 7th transformer layer.
> >The paper lacks discussion on the editing targets. As a LoRA-like method, ELoRA should be applicable to every module in the model (FFN or Attention), but the paper does not discuss whether there would be differences when various modules serve as editing targets.
>
>
> To address your question on module applicability, we conducted a new experiment on Qwen2.5-VL-7B for 500 edits, comparing edits on FFN modules versus Attention Projection modules (`q_proj`, `v_proj`). The results are shown below.
> | Edited module | Reliability | T-Gen | M-Gen | T-Loc | M-Loc | Real. Avg |
> | :--- | :---: | :---: | :---: | :---: | :---: | :---: |
> | FFN * | 98.60 | 88.40 | 89.00 | 74.20 | 86.00 | 87.24 |
> | Projection | 98.80 | 69.00 | 39.00 | **89.60** | **95.80** | 78.44 |
> | FFN + Projection | **99.00** | **88.60** | **89.60** | 77.60 | 86.60 | **88.28** |
>
> \* denotes the default setting in our evaluation.
>
> While editing Attention Projection yields excellent locality, it suffers significantly in generality (e.g., M-Gen drops to 39.00%). FFN layers provide a much better balance. This aligns with the hypothesis that FFNs store factual knowledge while Attention heads manage information routing [1]. Notably, combining both yields the highest overall performance (88.28%), an encouraging result that demonstrates ELoRA's extensibility beyond FFNs.
>
> >What r is used in ELoRA?
>
> We also evaluated ELoRA with varying ranks ($r \in \{64, 128, 256\}$) on Qwen2.5-VL-7B, the results are shown below:
>
> | rank $r$ | Reliability | T-Gen | M-Gen | T-Loc | M-Loc | Real. Avg |
> | :--- | :---: | :---: | :---: | :---: | :---: | :---: |
> | 64 | **99.00** | 86.20 | 83.40 | **77.40** | **89.20** | 87.04 |
> | 128*| 98.60 | 88.40 | 89.00 | 74.20 | 86.00 | 87.24 |
> | 256 | 98.90 | **91.58** | **95.35** | 70.44 | 83.99 | **88.05** |
> \* denotes the default setting in our evaluation.
>
> Increasing $r$ improves generality (due to a larger capacity to capture the knowledge manifold) but slightly trades off locality. We selected $r=128$ as the optimal balance for our reported results.
>
> We have included these analyses in the revised Appendix E to provide a reference for future work.
>
> ## Reference
> [1] Knowledge Neurons in Pretrained Transformers, ACL, 2022.

---

> > ### Comment · Reviewer_og9B · 2025-11-27
> >
> > I consider this to be a solid piece of work. Among the edited papers I have reviewed from the same period, it reaches a level that warrants acceptance (20%-25% poster).
> >
> > In my view, its main innovation lies in combining the respective advantages of LoRA and key-value pairs—an idea I had previously intended to explore but did not have the time to pursue. Additionally, the extension of editing to other modalities, despite certain limitations, is something I am pleased to see, as it contributes to the broader application of editing techniques.
> >
> > Noting that two reviewers have still given it a score of 4, I have decided to raise my score to 7 to better reflect the paper's accurate contribution. However, as a score of 7 is not an available option, I am adjusting both my overall recommendation and confidence scores to 5 accordingly.
> >
> > Furthermore, I believe the current editing paradigm should move beyond the constraints of methods like MEMIT and LoRA. In fact, since the advent of ROME, there has been a lack of new paradigms to break the mold. A promising direction for in-depth exploration in knowledge editing is leveraging Reinforcement Learning (RL) and on-policy methods to enhance its generalization. Similarly, this is a line of thinking that the multimodal editing community should also consider. I am curious to know the authors' perspectives on this issue.

---

> > > ### Author Response · Authors · 2025-11-27
> > > **Thanks to Reviewer og9B and Follow-up Discussion**
> > >
> > > Dear Reviewer og9B,
> > >
> > > **Thank you sincerely for raising your score to 7 as a recognition of our work.**
> > >
> > > Your comments are supportive and motivating for us, especially your observation on the stagnation of current editing paradigms and the call for new directions. We are glad to hear that our work resonates with this line of thinking.
> > >
> > > During the development of our work, we found that LoRA-based editing provides flexible control over different editing targets in the low-rank space, while maintaining stronger reliability than locate-then-edit approaches. This observation motivated us to design ELoRA to better resolve the conflicting objectives in multimodal editing.
> > >
> > > We are also exploring other paradigms in knowledge editing. Recently, we noticed the learning-to-edit paradigm, which teaches LLMs the skill to apply updated knowledge into answers through SFT or RL. This line of work learns editing behavior from editing data. Such methods could preserve locality, as the editor learns skills but not the knowledge. The generality can also be improved by reward design. However, they still suffer from insufficient knowledge internalization, the same issue in in-context editing, which we consider a central goal of knowledge editing, and thus this direction requires further investigation.
> > >
> > > We agree that, on the one hand, the on-policy mechanism could take both historical edits and current model states into consideration, which is something that existing approaches rarely do. On the other hand, as you suggested, RL may lead to better generalization because it optimizes editing behavior through long-term objectives rather than relying on supervised signals.
> > >
> > > We wonder whether on-policy editing can benefit from ideas in in-context approaches. Modifying parameters inevitably brings catastrophic forgetting, while in-context methods avoid this outcome since no weights are changed, although the knowledge is not internalized and disappears without context. A possible direction is to update parameters with on-policy RL to improve generalization and avoid overfitting to a single edit. The model could then evaluate each edit by estimating its long-term credit (e.g. knowledge conflict sensitivity, generality). Edits with low credit may remain as in-context knowledge, while high-credit edits could be internalized into parameters. We do not know the best way to estimate future credit, but developing such evaluation signals may be essential for reliable knowledge editing.
> > >
> > > **Thank you again for your recommendation of our work** and for broadening our understanding of where knowledge editing may progress.
> > >
> > > Best regards,
> > >
> > > Submission3972 Authors

---

### Official Review · Reviewer_QMCg · 2025-10-31

**Soundness:** 3
**Presentation:** 4
**Contribution:** 3
**Rating:** 6
**Confidence:** 4

**Summary:**

This paper introduces ELoRA, a novel knowledge editing framework that enables precise and generalizable updates. The method addresses a key challenge that existing approaches struggle to balance knowledge preservation and new knowledge generalization in multimodal contexts. Build upon LoRA, ELoRA decomposes LoRA updates into two subspaces: (1) a null space for preserving knowledge, and (2) a knowledge space for promoting generalizable updates. ELoRA achieves both high locality and high generality with this subspace decomposition. Experiments demonstrate consistent gains over standard LoRA-based methods across reliability, generality, and locality. Overall, this paper makes a technically sound and empirically validated contribution by providing a framework that balances knowledge preservation and semantic generalization in multimodal knowledge editing.

**Strengths:**

+ The combination of subspace decomposition and LoRA fine-tuning is creative in multimodal knowledge editing.
+ The theoretical analysis of null space and knowledge space constructions are well-motivated, supported by clear mathematical formulations. The experiments are comprehensive, covering both triplet-structured knowledge and free-form real-world multimodal knowledge. The performance gains are consistent across multiple models.
+ The paper is well-organized and clearly written. The figures are well-designed.
+ The work is impactful for both the knowledge editing and parameter-efficient fine-tuning. It offers a general framework to update multimodal models.

**Weaknesses:**

+ While the authors claim that the target knowledge is encoded within a low-dimensional, invariant manifold in the model's internal representations, this assumption appears to be empirically motivated rather than theoretically or experimentally substantiated. This hypothesis is conceptually appealing but lacks rigorous analytical or empirical verification.
+ The paper acknowledges that constructing the projection matrix for the knowledge space requires approximately 45 seconds per edit when using 511 perturbed samples, which is a nontrivial cost for real-time or batch editing applications. Although the authors claim that this cost can be linearly reduced by decreasing the number of perturbed samples, the paper does not analyze how the number of perturbed samples affects editing performance, making the efficiency-performance trade-off unclear.
+ The authors highlight the necessity of applying separate perturbations for visual and textual modalities; however, the paper does not clearly specify how the covariance matrix from the two modalities are fused or weighted when constructing the knowledge space. Is the overall covariance matrix obtained through direct concatenation or another aggregation strategy? If so, how does the method ensure that the fusion process balances different modalities to prevent one modality from dominating the overall subspace representation?
+ The paper does not clarify how the text-image pairs used to construct the null space are selected. Since the null space is intended to represent preserved knowledge, the sampling strategy could affect its coverage. Without specifying selection criteria, or sensitivity analysis, it is difficult to assess the stability of the null space.
+ The paper reports locality scores but lacks an analysis of the edited models' downstream task performance, which would help quantify the broader impact of editing on the models' general capabilities.

**Questions:**

Please see the above Weaknesses.

---

> ### Author Response · Authors · 2025-11-21
> **Response to Reviewer QMCg (Part 1)**
>
> Thank you for your effort and positive assessment of our creative idea, clear motivation, comprehensive experiments, and clear writing. In response to your major concerns, we provide detailed clarifications below.
>
> ## On the assumption of a low-dimensional invariant manifold.
>
> > While the authors claim that the target knowledge is encoded within a low-dimensional, invariant manifold in the model's internal representations, this assumption appears to be empirically motivated rather than theoretically or experimentally substantiated. This hypothesis is conceptually appealing but lacks rigorous analytical or empirical verification.
>
> Thank you for this good question. We agree that this is an empirically motivated hypothesis, and it is indeed a central topic of active research in the mechanistic interpretability community.
>
> This assumption is well-founded: it directly builds upon the **linear representation hypothesis** [1], which has found strong empirical support in prior knowledge editing work. For example, ROME [2] successfully edits atomic facts via **rank-one** (i.e., 1-dimensional) updates, providing empirical evidence that simple facts lie in extremely low-dimensional, linear subspaces.
>
> Our assumption extends the linear representation hypothesis. As multimodal knowledge is more intricate than the simple unimodal triplets edited by ROME, we posit that it is encoded not in a 1D direction, but in a **low-dimensional manifold**. This view aligns with emerging interpretability research [3], which is formally analyzing "why and how a feature might be represented as a manifold". This research provides direct evidence that **features exist as "continuous, nonlinear" structures (e.g., "Years of the 20th century") and proposes a "multidimensional linear representation hypothesis" to account for this**.
>
> We have **clarified this theoretical grounding in Section 3.4** of the revised manuscript. While formally proving the general manifold hypothesis is a key research goal for the interpretability community, our work focuses on leveraging this principle to construct an effective editing framework.
>
> Our proposed knowledge space is a computational method for identifying the low-dimensional manifold. Ablation study in Table 2 provides the experimental validation for our approach: a significant gain in generality (e.g., M-Gen rising from 47.54% to 63.12%) is achieved when the knowledge space ($M_5$) is added, compared to the null-space-only baseline ($M_2$).
>
>
> ## On the computational cost and perturbation sample size.
>
> >The paper acknowledges that constructing the projection matrix for the knowledge space requires approximately 45 seconds per edit when using 511 perturbed samples, which is a nontrivial cost for real-time or batch editing applications. Although the authors claim that this cost can be linearly reduced by decreasing the number of perturbed samples, the paper does not analyze how the number of perturbed samples affects editing performance, making the efficiency-performance trade-off unclear.
>
> Thank you for highlighting the practical issue of computational cost. In multimodal editing, achieving reliable generality remains the primary bottleneck, as evidenced by Table 1. Our design prioritizes breaking this performance ceiling, viewing controllable overhead as a necessary trade-off for the generalized editing that prior methods lack.
>
> To quantify this trade-off, we conducted a new ablation study on the sample size $B$ (see table below). Reducing $B$ from 511 to 127 lowers the construction time to about 11 seconds (a 75% reduction), while the Real-world Average score remains highly robust (about 87%). Even at $B=63$ (about 6 seconds), the performance is comparable to the peak setting. This demonstrates that the reported 45-second cost is an upper bound, offering significant flexibility for time-sensitive applications without sacrificing performance.
>
> Furthermore, compared to explicit distribution generation (e.g., using external diffusion models or LLMs to generate editing samples), ELoRA's implicit approximation via internal state perturbation is computationally more efficient and self-contained. We have added this analysis to Appendix E.1 in the revised manuscript.
>
> | Perturbed sample size | Reliability | T-Gen | M-Gen | T-Loc | M-Loc | Real. Avg |
> | :---: | :---: | :---: | :---: | :---: | :---: | :---: |
> | 63 | 98.69 | 88.43 | 89.48 | 73.97 | 85.76 | 87.26 |
> | 127 | **99.00** | **89.00** | 89.00 | 74.00 | 84.20 | 87.04 |
> | 255 | 98.80 | 88.60 | **89.80** | 75.00 | 85.00 | 87.44 |
> | 511* | 98.60 | 88.40 | 89.00 | 74.20 | 86.00 | 87.24 |
> | 1023 | 98.69 | 88.43 | 89.48 | **75.72** | **86.20** | **87.70** |
>
> \* denotes the default setting in our evaluation.

---

> ### Author Response · Authors · 2025-11-21
> **Response to Reviewer QMCg (Part 2)**
>
> ## Regarding the fusion strategy for multimodal covariance matrices.
>
> > The authors highlight the necessity of applying separate perturbations for visual and textual modalities; however, the paper does not clearly specify how the covariance matrix from the two modalities are fused or weighted when constructing the knowledge space. Is the overall covariance matrix obtained through direct concatenation or another aggregation strategy? If so, how does the method ensure that the fusion process balances different modalities to prevent one modality from dominating the overall subspace representation?
>
> Sorry for causing confusion. We do not fuse or weight two separate covariance matrices. Instead, we generate a unified set of $B$ perturbed samples that includes both visual and textual perturbations, compute a covariance matrix $C_b$ for each perturbed sample, and then obtain the final covariance matrix $C_{avg}$ by averaging all retained $C_b$.
>
> This unified strategy is further supported by the visualization in **Figure 4b**. Although the activations from text and visual perturbations naturally form two sub-clusters, the PCA direction extracted from all perturbed samples defines a single axis that spans both modalities. This indicates that the knowledge space is constructed from a shared underlying direction, rather than from the fusion of modality-specific covariance.
>
> ## On the sampling strategy for constructing the null space.
>
> > The paper does not clarify how the text-image pairs used to construct the null space are selected. Since the null space is intended to represent preserved knowledge, the sampling strategy could affect its coverage. Without specifying selection criteria, or sensitivity analysis, it is difficult to assess the stability of the null space.
>
> Thank you for your insightful question. We agree that the sampling strategy used to construct the null space may influence the coverage of preserved knowledge, and we clarify our setup below.
>
> As shown in Figure 2, we incorporate textual and visual QA pairs (TQA and VQA), along with image-caption data, to represent the model’s pretrained world knowledge. Concretely, for the textual TQA component, we randomly sample 2,048 examples from NQ-Open [4] with a maximum input length of 1,024 tokens. For the multimodal component, we separately sample 512 examples from the E-VQA and E-IC training splits in the MMEdit benchmark, which are derived from VQAv2 [5] and COCO Caption [6], respectively. These three subsets jointly form the activation matrix used to construct the null space. We will add these details to the revised version.
>
> At present, the null space serves as a technique in continual learning, and a further investigation is needed to better understand how it can more effectively represent pretrained world knowledge. It should be noted that the core idea of our work lies in decomposing the editing space and generalizing edits through the proposed knowledge space. We adopt the standard random-sampling strategy in constructing the null space.
>
>
>
> ## On the downstream performance impact of knowledge editing.
>
> > The paper reports locality scores but lacks an analysis of the edited models' downstream task performance, which would help quantify the broader impact of editing on the models' general capabilities.
>
> To evaluate the edited model’s capabilities in general visual question answering, we conduct experiments on the MME[7] benchmark and Qwen2.5-VL-7B model, which provides a broad collection of multimodal tasks that measure a model’s capabilities. We average the performance of 100 edits.
>
> | Model            | MME (Perception) | MME (Cognition) |
> |------------------|----------------|---------------|
> | Qwen2.5-VL-7B (before editing)   | 1705    | 610      |
> | Qwen2.5-VL-7B (after editing)    |       1699         |    607            |
>
> ## Reference
> [1] The Linear Representation Hypothesis and the Geometry of Large Language Models, ICML, 2024.
>
> [2] Locating and Editing Factual Associations in GPT, NeurIPS, 2022.
>
> [3] The Origins of Representation Manifolds in Large Language Models, 2025.
>
> [4] Latent Retrieval for Weakly Supervised Open Domain Question Answering, ACL, 2019.
>
> [5] Making the V in VQA Matter: Elevating the Role of Image Understanding in Visual Question Answering, CVPR, 2017.
>
> [6] Microsoft coco captions: Data collection and evaluation server, 2015.
>
> [7] MME: A Comprehensive Evaluation Benchmark for Multimodal Large Language Models, 2023.

---

> > ### Comment · Reviewer_QMCg · 2025-11-26
> >
> > I appreciate the authors' detailed response and additional clarifications. I maintain my positive score.

---

> ### Author Response · Authors · 2025-11-26
>
> Dear Reviewer QMCg,
>
> Thank you for your follow-up comment and positive assessment. We appreciate your constructive feedback and are glad our clarifications were helpful.
>
> Best regards,
>
> Submission3972 Authors

---

### Official Review · Reviewer_56no · 2025-11-01

**Soundness:** 2
**Presentation:** 3
**Contribution:** 2
**Rating:** 4
**Confidence:** 3

**Summary:**

In this paper, the authors propose ELoRA, a novel solution that disentangles the conflicting editing objectives. Specifically, ELoRA decomposes the standard Low-Rank Adaptation (LoRA) update into two complementary subspaces: (1) a null space aligned with preserved knowledge, constructed via multimodal initialization to maintain the model’s general capabilities, and (2) a knowledge space extracted from the model’s internal states to multimodal perturbations, capturing invariant semantics of updates.

**Strengths:**

1. The authors propose ELoRA, a novel solution that disentangles the conflicting editing objectives. Specifically, ELoRA decomposes the standard Low-Rank Adaptation (LoRA) update into two complementary subspaces: (1) a null space aligned with preserved knowledge, constructed via multimodal initialization to maintain the model’s general capabilities, and (2) a knowledge space extracted from the model’s internal states to multimodal perturbations, capturing invariant semantics of updates.

2.  The authors conduct extensive experiments on various LMMs, including LLaVAv1.5-7B, Qwen2.5-VL-7B, and Phi-4-multimodal, show that ELoRA outperforms most LoRA-based methods by an average of 14.2% accuracy across three metrics:

**Weaknesses:**

1. The construction of the knowledge space is contingent on several key hyperparameters: the window length for visual masking, the noise variance (σ) for text, the similarity threshold, and the number of perturbed samples (B). Figure 3 and 4a show that performance is sensitive to these choices. However, the process for selecting the final values used in the main experiments (e.g., why a specific θ was chosen) is not adequately justified. A more systematic analysis or a principled strategy for setting these parameters is needed to ensure reproducibility and robustness.

2.​​ The perturbation strategies seem somewhat arbitrary. The paper would benefit from a deeper discussion on whythese specific perturbations are suitable for extracting "invariant semantics." Are there more principled, modality-specific augmentation techniques that could be more effective?

3. Some widely-used baselines, such as ROME、MEMIT, should be compared

**Questions:**

NA

---

> ### Author Response · Authors · 2025-11-21
> **Response to Reviewer 56no (Part 1)**
>
> Thank you for your valuable feedback and for recognizing our novelty and comprehensive experiments. We are committed to addressing each concern you have raised.
>
> ## Regarding hyperparameter choices in knowledge space construction.
>
> > The construction of the knowledge space is contingent on several key hyperparameters: the window length for visual masking, the noise variance (σ) for text, the similarity threshold, and the number of perturbed samples (B). Figure 3 and 4a show that performance is sensitive to these choices. However, the process for selecting the final values used in the main experiments (e.g., why a specific θ was chosen) is not adequately justified. A more systematic analysis or a principled strategy for setting these parameters is needed to ensure reproducibility and robustness.
>
> Thank you for your question. **First, we would like to clarify** that all key hyperparameters in the proposed ELoRA were determined through ablation studies, and the final choices in our main experiments are directly supported by the ablation results provided in **Figure 3 and Figure 4a**. **Second, we provide additional justifications** for our choices for the similarity threshold $\theta$, the window length $w_l$, the noise variance $\sigma$, and the number of perturbed samples $B$.
>
> - **Similarity threshold $\theta$** determines the extent of perturbations included when constructing the knowledge space, striking a balance between enriching diversity and maintaining invariant semantics. Results in Figure 4a (right) show that as the similarity threshold increases, the generality initially rises (due to discarding stronger perturbations of both textual and visual modalities) and then drops (due to degrading diversity and a tendency to overfitting to sample-specific noise), indicating that moderate diversity is beneficial, while both insufficient and excessive perturbation can harm generality. Accordingly, in our evaluations, we fix the threshold at 0.05, as it corresponds to the peak performance observed in Figure 4a (right).
> - **Window length $w_l$** and **noise variance $\sigma$** jointly shape the distribution of perturbed samples, aiming to achieve rich yet semantically consistent variations when constructing the knowledge space. Our choices are informed by the systematic analysis in Figure 3: **(1) Rich variation** requires that the distribution not collapse toward a single peak near $\theta=1$. However, when $\sigma = 0$ or $0.001$, the distribution lacks mid-range similarities (e.g., from 0.4 to 0.6) and is substantially shifted to $\theta=1$. **(2) Semantic consistency** requires that the distribution retain sufficient samples with similarity above 0.5. However, when $\sigma = 0.05$, $0.1$ or $0.5$, the distribution shifts toward $\theta=0$, resulting a near absence of samples with $\theta>=0.5$. In contrast, $\sigma = 0.01$ enriches the diversity and preserves the semantics, exhibiting a peak distribution near $\theta=0.5$. Furthermore, as evidenced in Figure 4a (left), varying the window length has minimal impact on overall performance.
> - **Perturbed samples $B$** affect the number of covariance matrices computed in the knowledge space. We additionally conducted a quantitative analysis across different $B$ on Qwen2.5-VL-7B for 500 edits in E-VQA. The results below show that the model remains robust with only marginal performance variation. We have added the results in Appendix E.1.
>
>     | Perturbed sample size | Reliability | T-Gen | M-Gen | T-Loc | M-Loc | Real. Avg |
>     | :---: | :---: | :---: | :---: | :---: | :---: | :---: |
>     | 63 | 98.69 | 88.43 | 89.48 | 73.97 | 85.76 | 87.26 |
>     | 127 | **99.00** | **89.00** | 89.00 | 74.00 | 84.20 | 87.04 |
>     | 255 | 98.80 | 88.60 | **89.80** | 75.00 | 85.00 | 87.44 |
>     | 511* | 98.60 | 88.40 | 89.00 | 74.20 | 86.00 | 87.24 |
>     | 1023 | 98.69 | 88.43 | 89.48 | **75.72** | **86.20** | **87.70** |
>
>     \* denotes the default setting in our evaluation.
>
>
> Overall, our choices are guided by systematic and quantitative analysis. We will release our code to further support reproducibility.

---

> ### Author Response · Authors · 2025-11-21
> **Response to Reviewer 56no (Part 2)**
>
> ## Why can the perturbation strategies extract invariant semantics?
>
> > The perturbation strategies seem somewhat arbitrary. The paper would benefit from a deeper discussion on whythese specific perturbations are suitable for extracting "invariant semantics." Are there more principled, modality-specific augmentation techniques that could be more effective?
>
> The reviewer’s concern about perturbation strategies helps advance a deeper discussion of how we extract invariant semantics when constructing the knowledge space. **To address this concern, we first provide clarification and then conduct an interpretability analysis.**
>
> We employ a principled filtering mechanism (Eq. 9) with a predefined similarity threshold $\theta$ to select aligned internal states $\tilde{K}_b$ and ensure semantic control. Perturbations that exceed this threshold are discarded and considered to introduce semantic changes, which may compromise editing generality. This mechanism strikes a balance between leveraging reliable generality introduced by arbitrary perturbations and preserving invariant semantics through filtering.
>
> Furthermore, we conduct **an interpretability analysis** on the Qwen2.5-VL-7B model and E-VQA dataset to demonstrate that our perturbation strategy with filtering is effective for extracting modality-specific invariant semantics. **(1) For the textual modality**, we map perturbed textual embeddings back to tokens by multiplying them with the transpose of the embedding weight matrix and taking the argmax, and then decode the tokens into texts. We assess semantic variance between perturbed inputs $\{x_b\}$ and the original input $x_0$ using classical BLEU-4, ROUGE-L, and BERT-based models (e.g., Sentence-BERT [1]) to extract sentence embeddings. We also measure token-level differences using the proportion of identical tokens. The results are as follows. We partition the perturbations according to the predefined similarity threshold $\theta=0.05$.
>
> |               | $\theta \leq 0.05$ | $\theta > 0.05$ |
> | ------------- | :------------------: | :---------------: |
> | BLEU-4 (%)       |        18.88            |    40.16            |
> | ROUGE-L (%)      |      49.14              |    64.68          |
> | Sentence-BERT (%)  |      83.76            |     93.96          |
> | Token-level Consistency (%)         |  65.36                  |      87.32           |
>
> We observe a clear gap across all semantic metrics between perturbations with similarity scores above 0.05 and those below it, confirming that the perturbations with smaller similarity scores reflect severe semantic changes, and are therefore discarded to preserve invariant semantics in the textual modality.
>
> **(2) For the visual modality**, we visualize the impact of perturbed visual embeddings on the image as shown in **the newly added Figure 5 in the revised manuscript**. To map the masked visual projected embeddings $\mathbf{Z}$ (denoted in Appendix C.2) back to the image, we adopt a gradient-weighted perturbation mapping similar to Grad-CAM [2]. Specifically, we attribute the L2 loss between the perturbed embeddings $\{\mathbf{Z}_b\}$ and the original embedding $\mathbf{Z}_0$ to the $24\times24$ visual patches via gradients and then interpolate the resulting attribution map back to the original image resolution $336\times336$. The visualization shows that low-similarity perturbations reflect cases where key visual subjects are disrupted (e.g., tennis racket and tennis ball), thereby hindering the underlying semantics related to the editing subjects (e.g., how many tennis balls are in the picture?).
>
> Based on the above analysis, we demonstrate that our arbitrary perturbation strategy with filtering is simple yet effective for extracting invariant semantics in both textual and visual modalities. Although other augmentation techniques may be feasible, such as synonym substitution or image generation, they typically incur higher computational cost. We have added modality-specific interpretability analysis of perturbations in Section 4.3 in the revised manuscript.

---

> ### Author Response · Authors · 2025-11-21
> **Response to Reviewer 56no (Part 3)**
>
> ## Comparison with ROME and MEMIT.
>
> > Some widely-used baselines, such as ROME、MEMIT, should be compared.
>
> Thank you for your advice. We have adapted ROME [3] and MEMIT [4] for multimodal knowledge editing in the E-VQA benchmark. Since subject tokens required for ROME and MEMIT are not explicitly provided in the E-VQA dataset, we treat the last token of the prompt as the subject token for knowledge updates. The real-world editing performance on LLaVA-v1.5-7B is summarized below.
>
> | Methods   | Reliability | T-Gen | M-Gen | T-Loc | M-Loc | Real. Avg |
> |-----------|-------------|-------|-------|-------|-------|-----------|
> | ROME      | 81.27       | 59.53 | 59.29 | 85.19 | **91.35** |  75.33    |
> | MEMIT     | 71.09       | 69.42 | 60.25 | 77.50 | 87.53 |  73.15    |
> | ELoRA    |  **93.74**      | **80.46** | **63.12** | **97.09** |84.90  | **83.86**     |
>
>
> From the results, ELoRA demonstrates comprehensive superiority over both ROME and MEMIT. We have incorporated this comparison into the revised manuscript (please see Table 1).
>
> ## Reference
> [1] Sentence-BERT: Sentence embeddings using siamese bert-networks, 2019.
>
> [2] Grad-CAM: Visual explanations from deep networks via gradient-based localization, ICCV, 2017.
>
> [3] Locating and editing factual associations in GPT, NeurIPS, 2022.
>
> [4] Mass-editing memory in a transformer, ICLR, 2023.

---

> ### Author Response · Authors · 2025-11-27
> **A Gentle Reminder as We Approach the Final Week.**
>
> Dear Reviewer 56no，
>
> Thank you again for your valuable time and constructive feedback.
>
> As the rebuttal period is nearing its end, we would like to kindly check whether our previous response has addressed your main concerns. If there are any aspects that you feel need further clarification or improvement, please let us know, and we would be more than happy to provide additional explanations or revisions.
>
> We sincerely appreciate your thoughtful review and support.
>
> Best regards,
>
> Submission3972 Authors

---

### Author Response · Authors · 2025-11-23
**General Response**

## General Response
We thank all reviewers for their thoughtful and constructive feedback.

We are glad that reviewers (**R-56no**, **R-QMCg**, **R-og9B**) recognize the novelty and effectiveness of the proposed ELoRA, which disentangles the conflicting editing objectives in multimodal knowledge editing, and appreciate the contribution our work makes towards advancing the community:

- "The authors propose ELoRA, a novel solution that disentangles the conflicting editing objectives..." **R-56no**
- "The combination of subspace decomposition and LoRA fine-tuning is creative in multimodal knowledge editing." **R-QMCg**
- "The work is impactful for both the knowledge editing and parameter-efficient fine-tuning." **R-QMCg**
- "The authors effectively combined the advantages of editing methods and LoRA methods." **R-og9B**

We are glad that reviewers (**R-56no**, **R-QMCg**) find the experimental results convincing and thorough:

- "The authors conduct extensive experiments on various LMMs, including LLaVAv1.5-7B, Qwen2.5-VL-7B, and Phi-4-multimodal..." **R-56no**
- "The experiments are comprehensive, ... The performance gains are consistent across multiple models." **R-QMCg**

We also thank for reviewers (**R-QMCg**, **R-92Un**) appreciating our clear and well-organized writing:

- "The paper is well-organized and clearly written. The figures are well-designed." **R-QMCg**
- "The paper is clearly written and well-organized, with fluent language that makes the presentation easy to follow." **R-92Un**

## Updates and new experiments

We summarize the new experiments and discussions inspired by the reviewer's comments below.

- We address the concerns from **R-og9B** and **R-92Un** regarding our contribution by (1) clarifying that the complementary knowledge space is an essential design, supported by theoretical analysis (i.e., low-dimensional invariant manifold recognized by **R-QMCg**), and (2) demonstrating through the comprehensive results in Table 1 that neither purely null-space methods nor unimodal editing methods are sufficient to resolve the unique “trilemma’’ in multimodal knowledge editing.
- We respond to the concerns from **R-56no**, **R-QMCg**, and **R-92Un** regarding the rationale of perturbation strategies by incorporating **additional interpretability analyses for both textual and visual modality**, as presented in newly added Figure 5 and Appendix F in the revised manuscript. We further provide a systematic analysis of hyperparameter selections used in constructing the knowledge space based on the ablation studies shown in Figure 3 and Figure 4a.
- We include new ablation studies requested by reviewers (**R-QMCg**, **R-og9B**) on perturbed sample size, editing targets, and LoRA rank, and we further provide trend analyses to clarify their effects. We have added them in Appendix E in the revised manuscript.
- We add experiments requested by reviewers (**R-56no** and **R-QMCg**): (1) an evaluation of ROME and MEMIT under multimodal editing (see Table 1), and (2) an assessment of the general performance impact after editing.

---

### Author Response · Authors · 2025-11-29
**Summary for AC Consideration**

Dear Area Chair,

We sincerely appreciate your time and effort in handling our submission.

During the rebuttal, we actively addressed the reviewers’ concerns through additional experiments and analyses. The corresponding clarifications and new experiments are summarized in our **General Response comment below** for your convenience.

Before any information leakage occurred, **Reviewer QMCg** maintained their positive score ($6 \to 6$), and **Reviewer og9B** commented that a score of 7 would best reflect their assessment of our contribution ($6 \to 7$). Since such a score was not an available option, the reviewer instead assigned the highest recommendation and confidence scores and recommended acceptance in their comment.

As we have not received responses from **Reviewer 56no** and **Reviewer 92Un** in the discussion period, we are fully prepared to offer any clarification or additional information that may assist you at your convenience.

Thank you again for your consideration.

Best regards,

Submission3972 Authors

---

### Meta-Review · Area_Chair_g3v9 · 2026-01-07

**Summary:**

This paper proposes ELoRA that disentangles the conflicting editing objectives for simultaneously preserving pre-trained knowledge and generalizing new knowledge in intricate multi-modal contexts. In specific, ELoRA decomposes LoRA updates into two subspaces: (1) a null space for preserving knowledge, and (2) a knowledge space for promoting generalized updates. The experiments on several LLMs show that ELoRA have better reliability, generality and locality.

According to reviews, the most concerns are as follow.
1) The novelty is limited. The differences between the proposed AlphaEdit, LoRA-Null, and CorDA are not very clear.
2) The performance of some baselines on MMKE-Bench should be explained further?
3) Lack of some baselines.

**Reviewer Concerns:**

The novelty is limited. The proposed method is similar and extend to AlphaEdit.

**Reviewer Scores:**

According to the rebuttals, the final scores of reviewers are 6 7 4 4

---

### Decision · Program_Chairs · 2026-01-26

Reject